# UNISCALE: Adaptive Unified Inference Scaling via Online Joint Optimization of Model Routing and Test-Time Scaling

Kaiyu Huang [1,2]   Xingyu Wang [3]   Mingze Kong [3]   Zhubo Shi [1]   Yuqian Hou [4]
Hong Xu [5]   Zhongxiang Dai [2,3]   Minchen Yu [2,3]   Qingjiang Shi [1,2]

## Abstract

In real-world deployments of large language models (LLMs), balancing inference quality and computational cost has become a central challenge. Existing approaches tackle this trade-off along two largely independent dimensions: model routing, which switches among models of different scales to match request complexity, and test-time scaling (TTS), which adjusts inference-time compute within a fixed model for fine-grained control. However, this decoupled design introduces inherent limitations. Model routing yields coarse-grained, discrete performance changes due to the sparse set of model scales, while single-model TTS often encounters capacity ceilings and exhibits diminishing returns as compute increases. Moreover, treating the two mechanisms separately restricts adaptability in dynamic inference environments. To overcome these limitations, we introduce *Unified Inference Scaling (UIS)*, which unifies model routing and TTS in a single optimization space. Building on this formulation, we propose UNISCALE, an online framework that models adaptive UIS as a contextual multi-armed bandit problem and learns inference policies via LinUCB. The framework incorporates efficiency-aware learning and cost modeling to ensure stable and scalable optimization over high-dimensional action spaces. Evaluation shows that UNISCALE effectively exploits the synergy in the UIS space to deliver a fine-grained and consistently better

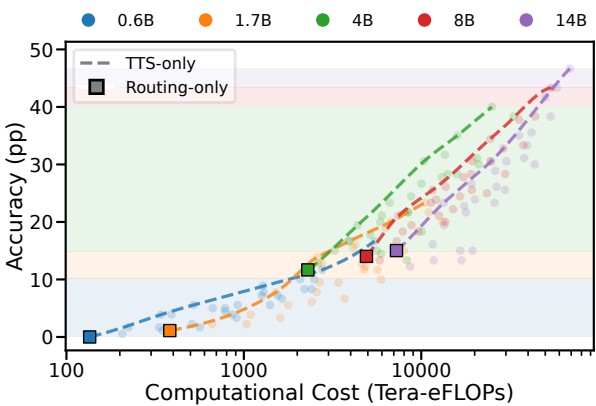

*Figure 1.* Accuracy-cost trade-offs under the UIS space. Squares and dashed lines represent routing nodes and single-model TTS trajectories, respectively. By jointly optimizing across both dimensions, UIS enables an expressive quality–cost frontier.

quality–cost trade-off across diverse, dynamic inference scenarios.

## 1. Introduction

In recent years, Large Language Models (LLMs) have demonstrated remarkable success across various tasks including complex reasoning, decision making, and multi-step problem solving (Guo et al., 2025; Jaech et al., 2024; Team, 2025). Real-world LLM applications span a wide spectrum of task difficulty, and meeting these requirements often demands substantial model capacity. However, higher capability typically incurs greater computational cost—either by invoking larger models with higher per-token latency and memory demands, or by increasing inference-time computation (e.g., longer decoding or additional test-time strategies). These increased costs can raise response latency and resource consumption, which is particularly consequential for interactive and large-scale online services. Therefore, real-world LLM deployments must navigate a trade-off between inference quality and computational cost.

To balance quality and cost, existing approaches typically

[1]School of Computer Science and Technology, Tongji University [2]Shenzhen Research Institute of Big Data, The Chinese University of Hong Kong, Shenzhen [3]School of Data Science, The Chinese University of Hong Kong, Shenzhen [4]College of Information Science and Electronic Engineering, Zhejiang University [5]Department of Computer Science and Engineering, Chinese University of Hong Kong. Correspondence to: Zhongxiang Dai <daizhongxiang@cuhk.edu.cn>, Minchen Yu <yuminchen@cuhk.edu.cn>, Qingjiang Shi <shiqj@tongji.edu.cn>.

*Proceedings of the 43rd International Conference on Machine Learning*, Seoul, South Korea. PMLR 306, 2026. Copyright 2026 by the author(s).

explore two largely independent dimensions, as shown in Figure 1. *Model routing* methods (Feng et al., 2025) select among models of different scales or capabilities, covering a broad quality–cost spectrum; however, they operate at a coarse granularity, as switching models induces discrete changes in both accuracy and cost. In contrast, *Test-Time Scaling (TTS)* methods (Snell et al., 2025) dynamically adapt the inference procedure of a given model at run time and provide fine-grained control, but they remain fundamentally constrained by the model's intrinsic capacity. Moreover, these two categories are often designed and configured in isolation, limiting their ability to operate in tandem in dynamic inference environments.

To achieve optimal quality–cost trade-offs in practice, an ideal approach should satisfy three requirements. First, it should offer a broad optimization space spanning models of different scales, enabling it to handle queries with diverse difficulty and computational demands (i.e., *broad coverage*). Second, it should provide fine-grained control over the inference procedure to realize precise quality–cost trade-offs (i.e., *fine granularity*). Finally, because online deployments face shifting query distributions, user objectives, and model availability (i.e., *environmental drift*), the approach should be able to continually adjust its inference strategy over time (i.e., *online adaptivity*).

To realize these requirements, we introduce *Unified Inference Scaling (UIS)*, an inference paradigm that treats model routing and TTS not as independent knobs, but as a *single, unified inference-time decision space*. Under UIS, inference is parameterized by a configuration that jointly specifies the base model and its associated TTS strategy (see Section 2.1). As illustrated in Figure 1, the resulting set of UIS configurations forms a rich design space in which routing and TTS interact: TTS can narrow performance gaps between discrete model scales, while routing to larger models when needed mitigates the diminishing returns of increasingly aggressive TTS on smaller models.

Building on this formulation, we propose UNISCALE, an online algorithm for solving the adaptive UIS problem. We formulate UIS configuration selection as an online contextual bandit task (Li et al., 2010), which naturally captures low-latency decision making under non-stationary environments (see Section 2.2). UNISCALE employs a Transformer-based encoder to extract query representations and uses the Linear Upper Confidence Bound (LinUCB) (Abbasi-Yadkori et al., 2011) algorithm to learn inference policies online (see Section 3), enabling continuous adaptation to environmental changes. Optimizing UIS in practice poses significant challenges due to the high-dimensional, heterogeneous configuration space and the stringent latency constraints of online inference. To ensure efficient and stable learning, we introduce three tightly integrated mechanisms (see

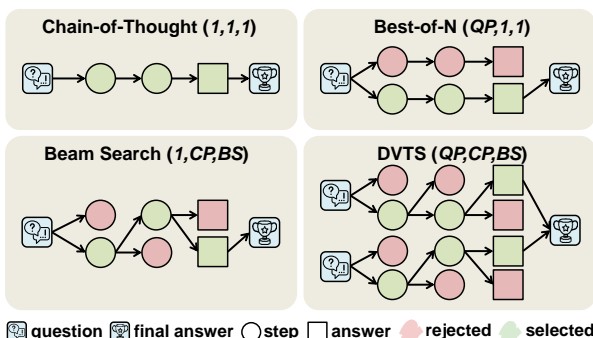

**question** **final answer** ◯ **step** □ **answer** ⬠ **rejected** ⬡ **selected**

*Figure 2.* Unified parameterization of TTS methods. Diverse TTS strategies are formalized via question parallelism ($QP$), candidate parallelism ($CP$), and beam size ($BS$).

Sections 3.3 and 3.4): (1) *Path-Aware Early Exiting* dynamically identifies and terminates low-potential inference paths to significantly reduce computational costs while guaranteeing inference quality, thereby optimizing the runtime performance of all UIS configurations; (2) *Dense Verification Feedback* densifies sparse binary correctness signals by incorporating native verifier scores from TTS, providing more accurate quality assessments to guide the system in selecting UIS configurations with superior performance; (3) *UIS Cost Model* utilizes equivalent FLOPs (eFLOPs) to map computational and memory overhead into a unified metric (Sadhukhan et al., 2025), ensuring consistency between the theoretical and actual costs of UIS configurations by providing accurate cost measurements.

Extensive experiments demonstrate that UNISCALE consistently outperforms baseline strategies across multiple settings, including TTS selection under fixed models (Section 4.1), model routing (Section 4.2), and full Unified Inference Scaling (Section 4.3). We further analyze UNISCALE via comprehensive ablation studies to clarify the contribution of each component (Section 5).

## 2. Background and Problem Setting

### 2.1. Foundations of Unified Inference Scaling

**Model Routing.** Model routing aims to dynamically select the most appropriate model $M$ from a heterogeneous pool of models based on the input query. However, while model routing methods enable coverage over a broad quality–cost range, they operate at a coarse granularity: switching between models often induces discrete and significant jumps in both inference quality and computational cost.

**Test-Time Scaling.** Test-Time Scaling (TTS) techniques enhance model reasoning by allocating additional inference-time computational budget (Wu et al., 2025). Utilizing Best-of-N (BoN) (Cobbe et al., 2021) or Process Reward

Model (PRM)-based search (e.g., Beam Search and DVTS), TTS methods refine reasoning via iterative verification and path evaluation (details in Section 3.3). As shown in Figure 2, we formalize these strategies into a three-dimensional parameter space: *Question Parallelism* ($QP$, the number of subtrees explored), *Candidate Parallelism* ($CP$, the number of parallel samples per step), and *Beam Size* ($BS$, the number of validated nodes retained). This parameterization allows for fine-grained control over quality–cost trade-offs, though it remains fundamentally upper-bounded by the base model's intrinsic capacity.

**Unified Inference Scaling.** To bridge these technical silos, we define Unified Inference Scaling (UIS) as a single unified inference-time decision space. We parameterize each inference execution through a joint configuration $(M, QP, CP, BS)$. This formulation constructs an expressive quality–cost frontier (see Figure 1) by bridging discrete model gaps with search intensity and surpassing search plateaus through model routing. This unified space enables inference strategies to be precisely tailored to the unique logical complexity and resource constraints of each query.

## 2.2. Adaptive Unified Inference Scaling via Bandits

To achieve online adaptivity under environmental drift, we formulate the UIS configuration selection as an online optimization problem through the lens of *contextual multi-armed bandits* (Li et al., 2010).

**Contextual Information.** For each arriving query $q_t$ at step $t$, the system observes a context vector $\mathbf{x}_t$. This vector serves as an abstract representation of the query's complexity, providing the necessary signal for the agent to estimate the potential utility of different UIS configurations.

**Unified Action Space.** We define a discretized action space $\mathcal{A}$, where each action $a \in \mathcal{A}$ corresponds to a specific UIS configuration $(M, QP, CP, BS)$. By treating routing and TTS as a joint action, UNISCALE can capture the cross-dimensional dependencies that independent methods ignore.

**Optimization Objective.** Upon executing action $a_t$, the agent receives a reward $r_t$, which is characterized by a joint function of inference quality and computational cost. The agent's goal is to minimize the cumulative regret:

$$R_T = \mathbb{E}\left[ \sum_{t=1}^{T} \left( r(q_t, a_t^*) - r(q_t, a_t) \right) \right], \qquad (1)$$

where $a_t^*$ is the optimal configuration for context $\mathbf{x}_t$. By minimizing this regret, the algorithm learns an adaptive policy that consistently selects the optimal UIS configuration tailored to the evolving environment.

---

**Algorithm 1** UNISCALE: Adaptive UIS via LinUCB
___________________________________________________
1: **Initialize:** $\mathbf{A}_0 \leftarrow \lambda\mathbf{I}$, $\mathbf{b}_0 \leftarrow \mathbf{0}$, $\mathbf{x}_{0,a_0} \leftarrow \mathbf{0}$, $r_0 \leftarrow 0$,
   action embeddings $\{\mathbf{s}_a\}_{a \in \mathcal{A}}$
2: **for** $t = 1$ to $T$ **do**
3:     Update $\mathbf{A}_t \leftarrow \mathbf{A}_{t-1} + \mathbf{x}_{t-1,a_{t-1}}\mathbf{x}_{t-1,a_{t-1}}^{\top}$
4:     Update $\mathbf{b}_t \leftarrow \mathbf{b}_{t-1} + r_{t-1}\mathbf{x}_{t-1,a_{t-1}}$
5:     Update $\hat{\boldsymbol{\theta}}_t \leftarrow \mathbf{A}_t^{-1}\mathbf{b}_t$
6:     Observe query $q_t$ and extract embedding $\mathbf{s}_{q_t}$
7:     **for all** $a \in \mathcal{A}$ **do**
8:         $\mathbf{x}_{t,a} \leftarrow \text{concat}(\mathbf{s}_{q_t}, \mathbf{s}_a)$
9:     **end for**
10:    Select UIS configuration

$$a_t = \arg\max_{a \in \mathcal{A}} \left( \hat{\boldsymbol{\theta}}_t^{\top}\mathbf{x}_{t,a} + \alpha\sqrt{\mathbf{x}_{t,a}^{\top}\mathbf{A}_t^{-1}\mathbf{x}_{t,a}} \right)$$

11:    Execute inference with configuration $a_t$
12:    Obtain reward $r_t$ according to Equation (4)
13: **end for**
___________________________________________________

## 3. The UNISCALE Framework

**Overview.** UNISCALE operates as an online closed-loop system (see Algorithm 1) designed to navigate the joint UIS decision space. In each iteration $t$, the system updates its reward estimator $\hat{\boldsymbol{\theta}}_t$ based on historical feature-reward pairs (Section 3.1). It then selects the optimal UIS configuration $a_t$ for the incoming query by maximizing the LinUCB acquisition function (Section 3.2). Following selection, UNIS-CALE executes the inference procedure on base model $M_t$, utilizing path-aware early exiting to optimize the runtime quality–cost trade-off (Section 3.3). Finally, the system evaluates the execution via dense verification feedback and the UIS cost model to yield a composite reward $r_t$ for continuous policy refinement (Section 3.4).

### 3.1. Updating the Reward Estimator

At the beginning of each iteration $t$, UNISCALE updates the reward estimator using the feedback $(\mathbf{x}_{t-1,a_{t-1}}, r_{t-1})$ observed in the previous round. We formulate the expected reward as a linear relationship $\hat{r}_t = \langle \mathbf{x}_{t,a_t}, \boldsymbol{\theta} \rangle$, where $\boldsymbol{\theta}$ is a learnable parameter vector shared across the entire UIS action space. To refine this estimator, the system incrementally updates the Gram matrix $\mathbf{A}_t$ and the feature-reward vector $\mathbf{b}_t$ to derive the current parameter estimate $\hat{\boldsymbol{\theta}}_t$, as detailed in Algorithm 1 (Lines 3–5). In practice, we employ the Sherman—Morrison formula to update $\mathbf{A}_t^{-1}$ via efficient rank-one updates. This avoids recomputing the inverse from scratch, reducing the per-round computational complexity from $\mathcal{O}(d^3)$ to $\mathcal{O}(d^2)$. This incremental mechanism ensures that the reward estimator continuously adapts to environmental drift with minimal computational overhead.

**Joint Semantic Representation.** To capture the intrinsic alignment between user requirements and system capabilities, UNISCALE maps queries and configurations into a shared latent space via a unified Transformer encoder. Specifically, we pre-compute the action semantic representations $\mathbf{s}_a$ based on the attribute descriptions of each UIS configuration (see Section B.2), which provides rich prior knowledge for online decision-making. During the inference phase, the same encoder maps the incoming query $q_t$ to its semantic embedding $\mathbf{s}_{q_t}$ in real-time. These two components are then concatenated to form the joint semantic representation $\mathbf{x}_{t,a} = [\mathbf{s}_{q_t}; \mathbf{s}_a]$, providing a high-dimensional grounding for accurate reward estimation.

**Justification of Linear Reward Modeling.** The motivation for adopting a linear estimator rather than a non-linear neural architecture is twofold: on the one hand, modern Transformer encoders have already captured the bulk of complex non-linear semantic structures during the feature extraction stage (Hu et al., 2024b), rendering a linear mapping sufficient to characterize the relationship between rewards and features; on the other hand, linear estimators offer significant advantages in terms of computational efficiency and theoretical tractability, making them highly suitable for online inference and frequent updates in dynamic system environments.

### 3.2. Selecting the Next Configuration $a_t$

Upon updating the reward estimator, we determine the optimal UIS configuration $a_t$ for the current query $q_t$ by maximizing the LinUCB acquisition function (Equation (2)). Specifically, UNISCALE utilizes the parameters $\hat{\boldsymbol{\theta}}_t$ of the current reward estimator to compute the acquisition values for each action within the candidate configuration set $\mathcal{A}$. The final UIS configuration $a_t$ is identified by selecting the action that yields the maximum value:

$$a_t = \arg\max_{a \in \mathcal{A}} \left( \mathbf{x}_{t,a}^\top \hat{\boldsymbol{\theta}}_t + \alpha \sqrt{\mathbf{x}_{t,a}^\top \mathbf{A}_t^{-1} \mathbf{x}_{t,a}} \right). \quad (2)$$

Within this expression, $\mathbf{x}_{t,a}^\top \hat{\boldsymbol{\theta}}_t$ denotes the predicted reward for configuration $a$ under the given context. The term $\sqrt{\mathbf{x}_{t,a}^\top \mathbf{A}_t^{-1} \mathbf{x}_{t,a}}$ serves as a principled measure of uncertainty regarding the estimated value, which is derived from the Gram matrix $\mathbf{A}_t$ accumulated from historical observations (refer to Section D.1 for details).

By tuning the hyperparameter $\alpha$, UNISCALE strikes a precise balance between two critical dimensions: (1) *Exploitation*: Leveraging historical observations to favor configurations with high predicted rewards. (2) *Exploration*: Encouraging an exhaustive search across the action space by prioritizing configurations with greater uncertainty. The synergy between reward estimation and principled exploration enables UNISCALE to converge rapidly within the

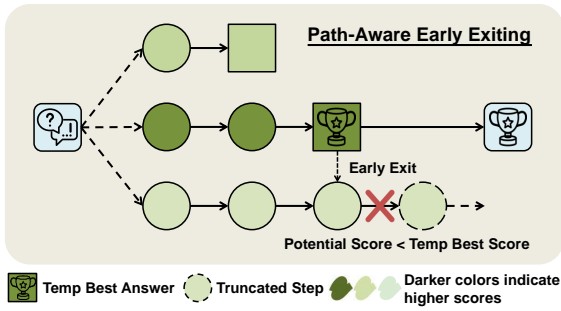

*Figure 3.* Illustration of the path-aware early exiting mechanism.

high-dimensional UIS decision space. This ensures the consistent selection of configurations on the optimal quality–cost frontier. We provide a detailed sensitivity analysis of the exploration factor $\alpha$ in Section C.3.

### 3.3. Executing Unified Inference Scaling

Upon determining the optimal UIS configuration $a_t = (M_t, QP_t, CP_t, BS_t)$, UNISCALE executes a parameterized TTS procedure on the target model $M_t$. This procedure is abstracted as a search forest consisting of $QP_t$ subtrees, which evolves through a cyclical iteration of the following four steps: (1) *State Generation*: For each expansion at step $j$, the system generates $CP_t$ intermediate inference states $s_{i,j,k}$ for every subtree. (2) *Process Verification*: Once all intermediate states for the current step are generated, a verifier assigns a score $v(s_{i,j,k})$ to each state. (3) *Path Evaluation*: An inference path $p_{i,j}$ of depth $j$ is defined as an ordered sequence of states within a subtree. Its cumulative score $V(p_{i,j})$ is calculated as the arithmetic mean of the scores of all states within the path: $V(p_{i,j}) = \frac{1}{j}\sum_{h=1}^{j} v(s_{i,h,k_h})$. (4) *Search Control*: The system ranks candidate branches based on their path scores. To manage computational complexity, only the top $BS_t$ paths in each subtree are retained for subsequent expansion. When a path meets the termination criteria (e.g., generating a complete answer or reaching the maximum depth), it is added to the set of completed paths $\mathcal{P}_{\text{done}}$. Finally, the system selects the result with the highest score as the final output: $y^* = \arg\max_{p \in \mathcal{P}_{\text{done}}} V(p)$.

**Inference Execution with Path-Aware Early Exiting.** To overcome the straggler effect inherent in traditional TTS workflows, where the system must wait for the slowest path to finish, UNISCALE introduces a path-aware early exiting mechanism (see Figure 3). The core of this mechanism lies in leveraging verifier scores as informative indicators of final answer correctness (detailed experiments in Section D.2) to perform real-time potential assessment of ongoing inference paths. During execution, the system dynamically maintains the current maximum score $V_{\text{max}}$ and the depth of the corresponding best path $D_{\text{best}}$ among all completed

paths: $V_{\max} = \max_{p \in \mathcal{P}_{\text{done}}} V(p)$. To improve efficiency, the maximum search depth $H_{\max}$ is dynamically updated as: $H_{\max} = \lceil \eta \cdot D_{\text{best}} \rceil$, where $\eta \geq 1$ is a tunable expansion factor. For any incomplete path $p_{i,j}$ with length $j$, the system determines its viability by calculating its theoretical maximum potential score: $\frac{j \cdot V(p_{i,j}) + (H_{\max} - j) \cdot v_{\text{sup}}}{H_{\max}} < V_{\max}$, where $v_{\text{sup}} = 1$ represents the theoretical upper bound of the verifier's score. If this condition is met, the system determines that even if the path performs perfectly in all subsequent steps, its final score cannot surpass the current best result. Consequently, the system terminates the expansion of that path immediately. By transforming path scores into online search control signals, this rule dynamically eliminates low-potential branches without altering the final selection criteria. This ensures that the inference cost of all UIS configurations is significantly reduced while maintaining their intrinsic reasoning quality, thereby optimizing the practical quality–cost frontier.

### 3.4. Evaluating the Selected Configuration $a_t$

In evaluating the selected UIS configuration $a_t$, UNISCALE constructs a composite reward function that precisely reflects the inference utility through a multi-dimensional metric system. The core logic of this function lies in the fine-grained characterization of inference quality and costs.

**Quality Assessment via Dense Verification Feedback.** To address the feedback sparsity of relying solely on binary answer correctness $\text{Correct}(a_t)$, UNISCALE introduces the *dense verification feedback* mechanism. This mechanism reshapes the quality assessment into a multi-dimensional evaluation framework, jointly driven by the binary correctness $\text{Correct}(a_t)$ of the final output $y^*$ and its intrinsic path score $V(y^*)$ (denoted as $\text{Score}(a_t)$). Specifically, as a continuous variable, the verifier score provides a denser supervision signal than binary labels, enabling the agent to perceive nuances in logical rigor across different reasoning paths. By capturing the evolution of these scores, the system can acutely identify latent progress during the reasoning process, thereby effectively guiding the agent to select better UIS configurations and achieve a final breakthrough in correctness. A systematic experimental analysis regarding the alignment between verifier scores and answer correctness is provided in Section D.2.

**Cost Measurement via UIS Cost Model.** To accurately quantify the cost of UIS configurations, we construct a *UIS cost model* $C_{\text{UIS}}$, which incorporates the prefill cost $C_{\text{prefill}}$, the incremental generation cost $C_{\text{inc}}^{(j)}$ of each inference step, and the corresponding verification cost $C_{\text{ver}}^{(j)}$ (see Section D.3 for details).

$$C_{\text{UIS}} = C_{\text{prefill}}(L_{\text{in}}) + \sum_{j=1}^{H} \left[ C_{\text{inc}}^{(j)} + C_{\text{ver}}^{(j)} \right]. \qquad (3)$$

*Table 1.* Default configuration range of the UIS space $\mathcal{A}$, utilizing the Qwen3 series (Yang et al., 2025) as candidate models and Skywork-o1-Open-PRM-Qwen-2.5-1.5B (He et al., 2024) as the verifier.

| Component | Configuration Details |
|---|---|
| Models | Qwen3 (0.6B, 1.7B, 4B, 8B, 14B, 32B) |
| TTS | $QP, CP \in \{2^0, \ldots, 2^6\}$, s.t. $QP \times CP \leq 2^6$ $BS = \{1, 2, 4\}$ for $CP \leq \{2^1, 2^3, 2^6\}$ |
| Verifier | Skywork-o1-Open-PRM-Qwen-2.5-1.5B |

Each cost component is calculated using equivalent FLOPs (eFLOPs) (Sadhukhan et al., 2025), which projects memory access pressure and computational load onto a unified dimension via hardware arithmetic intensity. This modeling enables $C_{\text{UIS}}$ to transcend the boundaries between compute-bound and memory-bound operations, providing an accurate characterization of the true cost of UIS on heterogeneous hardware. For subsequent reward computation, the system log-transforms the output $C_{\text{UIS}}(a_t)$ and applies min-max normalization to yield the standardized cost term $\tilde{C}_{\text{UIS}}(a_t)$.

**Composite Reward Function.** The final reward is a convex combination of quality and cost:

$$r_t = w_1 \cdot \text{Correct}(a_t) + w_2 \cdot \text{Score}(a_t) + w_3 \cdot (1 - \tilde{C}_{\text{UIS}}(a_t)). \qquad (4)$$

As users adjust reward weights $w_i$ to redefine priorities, UNISCALE promptly captures these shifts and identifies the optimal UIS configuration consistently.

## 4. Empirical Evaluation

We evaluate UNISCALE's online decision-making performance within a joint UIS space comprising multiple candidate models and diverse TTS strategies (see Table 1). The evaluation encompasses three deployment scenarios derived from practical requirements: optimizing TTS strategies for a specific model (Section 4.1), routing queries across distinct candidate models (Section 4.2), and performing full joint optimization within the complete UIS space (Section 4.3).

Our experiments are conducted on a total of 210 instances curated from the AIME'24 (Zhang & Math-AI, 2024), AIME'25 (Zhang & Math-AI, 2025), and MATH-500 (Aggarwal et al., 2023) datasets. Specifically, for MATH-500, we randomly sample 30 instances from each difficulty level (Levels 1–5). These 210 instances directly correspond to the experimental workflow, which consists of a 50-step warm-up followed by 160 policy-driven iterations. We compare UNISCALE against two categories of baselines: (1) Multi-armed Bandit baselines, including *Random* exploration, a *Greedy* strategy based on UNISCALE's reward estimator, *Thompson Sampling (TS)* (Chapelle & Li, 2011) that main-

*Table 2.* Main performance comparison across TTS , Routing , and UIS scenarios. Results represent the mean and standard deviation across five random seeds (3, 23, 42, 50, 57), excluding the 50-step warm-up phase. Performance is evaluated under Cost-Sensitive and Quality-Priority reward modes, with **best** and second-best results highlighted accordingly.

| Method | Metric | Cost-Sensitive | | | Quality-Priority | | |
|---|---|---|---|---|---|---|---|
| | | TTS | Routing | UIS | TTS | Routing | UIS |
| Random | Reward (↑) | 0.6937 (0.0052) | 0.4733 (0.0076) | 0.5731 (0.0143) | 0.5726 (0.0056) | 0.5055 (0.0095) | 0.6175 (0.0116) |
| | Accuracy (↑) | 42.12 (1.22) | 43.50 (1.29) | 53.00 (1.70) | 42.25 (1.09) | 43.25 (1.33) | 52.88 (1.79) |
| | Cost (↓) | 76.7 (9.7) | 2302.9 (190.6) | 1358.7 (220.1) | 73.2 (7.0) | 2370.4 (278.9) | 1359.8 (219.6) |
| Greedy | Reward (↑) | 0.7184 (0.0383) | 0.5873 (0.0406) | 0.5589 (0.0616) | 0.6184 (0.0435) | **0.5459 (0.0152)** | 0.5780 (0.0441) |
| | Accuracy (↑) | 43.75 (1.58) | 34.12 (6.63) | 45.00 (4.66) | 46.50 (2.46) | **50.00 (2.40)** | 52.88 (2.26) |
| | Cost (↓) | 54.0 (30.7) | 202.5 (145.9) | 660.6 (219.0) | 67.6 (44.4) | 2643.4 (916.7) | 3402.1 (1305.0) |
| MLP | Reward (↑) | 0.7006 (0.0486) | 0.6161 (0.0071) | 0.6301 (0.1032) | 0.5752 (0.0609) | 0.4963 (0.0124) | 0.6055 (0.0329) |
| | Accuracy (↑) | 41.75 (6.91) | 29.75 (2.08) | 47.00 (5.02) | 42.38 (5.18) | 42.00 (3.25) | 47.50 (4.95) |
| | Cost (↓) | 48.6 (20.4) | 154.7 (105.6) | 644.0 (769.8) | 46.1 (19.9) | 1807.3 (1238.6) | 353.4 (340.3) |
| k-NN | Reward (↑) | 0.6966 (0.0055) | 0.5819 (0.0047) | 0.6590 (0.0108) | 0.5146 (0.0107) | 0.5273 (0.0149) | 0.5807 (0.0180) |
| | Accuracy (↑) | 36.50 (1.51) | 34.62 (3.68) | 41.38 (2.83) | 36.25 (2.27) | 47.38 (1.99) | 46.75 (2.72) |
| | Cost (↓) | 49.0 (3.4) | 522.9 (65.6) | 326.0 (94.4) | 47.8 (4.5) | 2046.8 (327.9) | 1113.4 (253.0) |
| NeuralUCB | Reward (↑) | 0.6984 (0.0185) | 0.4849 (0.0194) | 0.5880 (0.0291) | 0.5580 (0.0087) | 0.4991 (0.0199) | 0.5929 (0.0132) |
| | Accuracy (↑) | 42.00 (2.14) | 42.62 (2.66) | 50.75 (1.50) | 41.12 (0.25) | 42.75 (3.66) | 50.25 (2.61) |
| | Cost (↓) | 70.6 (13.8) | 2216.6 (726.5) | 1558.2 (1120.7) | 68.4 (6.1) | 2679.1 (594.7) | 1819.7 (838.1) |
| TS | Reward (↑) | 0.7022 (0.0075) | 0.4541 (0.0086) | 0.5549 (0.0102) | 0.5901 (0.0089) | 0.5292 (0.0124) | 0.6243 (0.0050) |
| | Accuracy (↑) | 44.50 (1.55) | 48.37 (1.16) | 52.12 (2.11) | 44.88 (1.55) | 47.00 (2.00) | 54.00 (1.09) |
| | Cost (↓) | 67.9 (11.3) | 2992.7 (395.3) | 1762.8 (389.2) | 63.9 (8.4) | 3065.1 (495.0) | 1875.7 (396.7) |
| UNISCALE (ours) | Reward (↑) | **0.7535 (0.0071)** | **0.6196 (0.0039)** | **0.7079 (0.0110)** | **0.6450 (0.0265)** | 0.5347 (0.0183) | **0.6306 (0.0212)** |
| | Accuracy (↑) | 46.50 (1.02) | 29.00 (1.61) | 46.88 (1.72) | **48.12 (1.05)** | 47.87 (2.64) | **57.37 (1.94)** |
| | Cost (↓) | **23.3 (2.7)** | **82.9 (4.5)** | **49.4 (11.6)** | 26.6 (9.6) | 1924.6 (969.1) | 1374.7 (437.5) |
| Oracle | Reward (↑) | 0.8297 (0.0031) | 0.6226 (0.0044) | 0.8337 (0.0030) | 0.7426 (0.0047) | 0.6121 (0.0093) | 0.7924 (0.0048) |
| | Accuracy (↑) | 54.87 (0.92) | 37.50 (1.85) | 57.38 (0.92) | 59.12 (0.75) | 56.12 (2.07) | 68.12 (0.79) |
| | Cost (↓) | 10.5 (0.3) | 106.4 (6.1) | 10.6 (0.3) | 20.3 (0.7) | 1911.1 (86.4) | 115.7 (3.0) |

tains posterior reward distributions for exploration, *NeuralUCB* (Zhou et al., 2020) that utilizes neural networks for non-linear function approximation with optimistic exploration, and an *Oracle* that represents the theoretical performance upper bound. (2) Predictive Routing baselines inspired by RouterBench (Hu et al., 2024a), consisting of online *MLP* and *k-NN* routers that estimate performance based on historical observations. We additionally compare against BEST-Route (Ding et al., 2025) under compatible settings in Section B.5. To evaluate the algorithm's versatility, we conduct primary experiments under two practical reward modes, *Quality-Priority* and *Cost-Sensitive*. We also employ *Cost-Leaning* and *Quality-Leaning* variants solely for the Pareto analysis in Section 4.3.

We evaluate UNISCALE using two categories of metrics. For *static performance*, we report the mean Reward (see Equation (4)), Accuracy (in percentage points, pp), and Cost (in Tera-eFLOPs, see Section D.3) over the 160 policy-driven iterations. For *dynamic learning efficiency*, we track the cumulative Regret (see Equation (1)), Correct counts, and Cost (including the 50-step warm-up phase). The cumulative regret is further analyzed by defining Reg.@130 and Reg.@210 as the regret measured up to the 130th and 210th

steps, respectively.

Detailed experimental settings and results are provided in Sections B.1 to B.4. Additionally, we demonstrate UNIS-CALE's inherent task-agnostic generalizability through experiments across diverse tasks (Section B.6).

### 4.1. Scenario I: Adaptive Test-Time Scaling Selection

We evaluate the fine-grained quality–cost trade-offs by dynamically selecting configurations within the TTS subspace under a fixed Qwen3-0.6B capacity. Table 2 shows that while the TTS subspace offers sufficient flexibility to achieve high rewards in Cost-Sensitive mode (matching global UIS performance), a significant gap persists in Quality-Priority mode. This confirms that *the TTS subspace is ultimately limited by the base model capacity*, preventing the realization of wide-range quality–cost trade-offs. Notably, UNISCALE significantly outperforms all baseline methods across both reward modes, consistently securing the highest reward by identifying optimal configurations that yield the lowest inference cost in the Cost-Sensitive mode and the highest accuracy in the Quality-Priority mode.

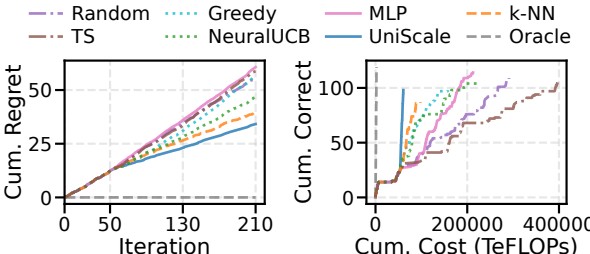

*Figure 4.* Cumulative regret and efficiency curves for UNISCALE and baselines in the Adaptive UIS scenario under Cost-Sensitive mode (including the 50-step warm-up phase).

### 4.2. Scenario II: Adaptive Model Routing

We evaluate the algorithm's capability for dynamic query routing across various model scales within the routing subspace under a fixed inference structure (e.g., CoT). The lower rewards compared to the global UIS space in Table 2 highlight the inherent limitations of the routing subspace, as it lacks the logical compensation provided by TTS and *suffers from performance jumps due to the sparsity of available model configurations*. While UNISCALE remains robust, its performance is comparable to the Greedy baseline. This reflects the low-dimensional nature of this routing subspace, where the exploration factor in UNISCALE incurs a slight penalty once the optimal model for the current distribution is rapidly identified during the warm-up phase. Crucially, as the Greedy strategy represents a special case of UNISCALE without exploration and relies on the same linear reward estimator, its competitive performance directly *validates the estimator's effectiveness* in accurately characterizing model capability boundaries and identifying query complexity.

### 4.3. Scenario III: Unified Inference Scaling

We evaluate the effectiveness of the UIS paradigm through the joint optimization of model and TTS selection within the global UIS space. Table 2 confirms that the UIS space yields the highest global rewards, validating the deep synergy between the two dimensions. Specifically, diverse TTS configurations smooth the discrete performance jumps of the routing subspace, while model switching breaks the capacity ceiling of single-model TTS. In this complex joint space, UNISCALE excels across both reward modes, consistently securing the highest rewards (and thus the minimum cumulative regret) by identifying configurations that yield the minimum cost in the Cost-Sensitive mode (see Figure 4) and the maximum accuracy in the Quality-Priority mode.

This trend is further illustrated in Figure 5, which presents the accuracy-cost frontier under different reward modes. Overall, UNISCALE consistently achieves a superior Pareto trade-off compared with all baselines, demonstrating its

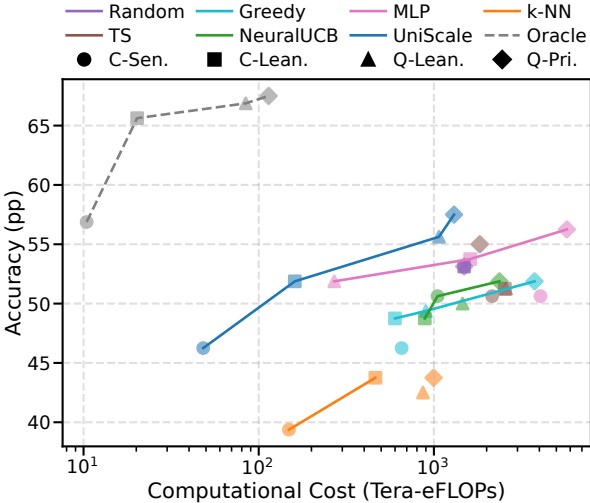

*Figure 5.* Accuracy-cost trade-offs of UNISCALE and baselines in the Adaptive UIS scenario. Distinct marker shapes represent four distinct reward modes: Cost-Sensitive (C-Sen.), Cost-Leaning (C-Lean.), Quality-Leaning (Q-Lean.), and Quality-Priority (Q-Pri.).

*Table 3.* Performance comparison between UNISCALE and a non-semantic baseline (*w/o Sem.*) across reward modes.

| Mode | Config. | Reg.@130 | Reg.@210 | Acc. | Cost |
|---|---|---|---|---|---|
| Q-Pri. | UNISCALE | **20.38** | **33.98** | **57.50** | 1303.3 |
| | *w/o Sem.* | 23.18 | 38.24 | 50.62 | **906.4** |
| C-Sen. | UNISCALE | **23.43** | **34.43** | 46.25 | **48.2** |
| | *w/o Sem.* | 26.74 | 41.56 | **46.88** | 629.7 |

ability to adaptively exploit the joint UIS space. Compared with existing baselines, it exhibits a steeper improvement trend, indicating more efficient configuration selection across diverse reward modes. Moreover, all variants of UNISCALE remain on the Pareto frontier across different reward weights, demonstrating strong robustness to coefficient variations while avoiding undesirable trade-offs between inference cost and generation quality. These results demonstrate that online joint optimization fully *unleashes the UIS paradigm's potential for wide-range, fine-grained trade-offs*.

## 5. Ablation Study

**Effectiveness of Action Semantic Representations.** We assess the impact of action semantic representations (see Section 3.1) on learning efficiency and decision quality by comparing UNISCALE against a baseline that represents actions as independent one-hot vectors (*w/o Sem.*). Table 3 shows that incorporating semantic features consistently yields lower cumulative regret across both reward modes. By evaluating results across different objectives, we find that UNISCALE facilitates a significantly broader

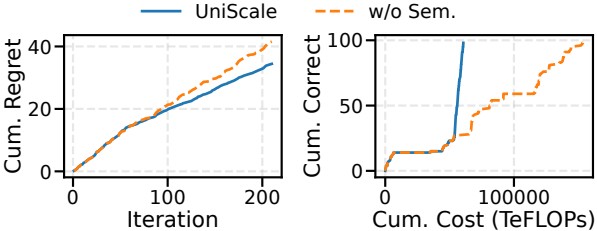

*Figure 6.* Cumulative regret and efficiency curves for UNISCALE and a non-semantic baseline under Cost-Sensitive mode.

*Table 4.* The impact of applying path-aware early exiting on accuracy, computational load (TFLOPs), memory access volume (TB), and inference cost.

| Configuration | Acc. | Comp. | Mem. | Cost |
|---|---|---|---|---|
| Standard TTS | 65.43 | 667.4 | 26.8 | 4851.4 |
| *w/ Early Exit* | 64.52 | 85.0 | 5.9 | 1002.9 |
| Change | -0.91pp | -87.26% | -78.06% | -79.33% |

range of quality–cost trade-offs: the accuracy difference between Quality-Priority and Cost-Sensitive modes reaches 11.25pp, with the latter's cost being only 3.7% of the former. In contrast, the baseline achieves a narrower accuracy gap of 4.37pp, with its Cost-Sensitive cost remaining at 69.5% of its Quality-Priority cost. These results demonstrate that action semantic representations effectively enhance UNIS-CALE's understanding of both inference quality and cost across diverse UIS configurations. By enabling cross-action knowledge transfer, this approach significantly accelerates policy optimization within high-dimensional search spaces. In addition to the Cost-Sensitive mode shown in Figure 4, we provide the full set of results in Section C.1.

**Efficiency Gains from Path-Aware Early Exiting.** We evaluate the impact of the path-aware early exiting mechanism (Section 3.3) by comparing the average performance across all UIS configurations. To balance exploration depth and pruning aggressiveness, we set the expansion factor $\eta = 1.2$. Table 4 shows that this strategy significantly optimizes inference efficiency by performing real-time potential assessment of ongoing inference paths. By dynamically eliminating redundant steps, the mechanism reduces computational load and memory access volume, which lowers the total inference cost with negligible impact on final accuracy. These findings confirm the efficacy of the mechanism in identifying logical convergence points and achieving high efficiency by pruning ineffective inference paths.

**Effectiveness of Dense Verification Feedback.** We examine the role of dense verification feedback (see Section 3.4) in guiding policy convergence compared to a baseline relying on sparse binary correctness (*w/o Dense Feedback*). Table 5 demonstrates that verification feedback provides a

*Table 5.* Effectiveness of dense verification feedback compared to binary correctness (*w/o Dense Feedback*) across reward modes.

| Mode | Configuration | Accuracy ($\uparrow$) | Cost ($\downarrow$) |
|---|---|---|---|
| Q-Pri. | UNISCALE | **57.50** | **1303.3** |
| | *w/o Dense Feedback* | 51.88 | 2674.9 |
| C-Sen. | UNISCALE | **46.25** | 48.2 |
| | *w/o Dense Feedback* | 43.75 | **47.4** |

denser supervision source, allowing the system to perceive subtle nuances in logical rigor. Specifically, UNISCALE achieves a 5.62pp accuracy improvement and a 51.3% reduction in inference cost in Quality-Priority mode, while achieving a 2.5pp accuracy gain with nearly identical computational expenditure in Cost-Sensitive mode. This validates that dense feedback effectively anticipates breakthroughs in final correctness, mitigating the reward sparsity inherent in traditional binary signals.

**Robustness to Non-stationary Environmental Drifts.** We evaluate the online adaptability of UNISCALE by simulating four non-stationary environments where environmental drifts are introduced at the 51st iteration, immediately following the warm-up phase. These encompass action space dynamics, involving the addition and removal (Add./Rem.) of 0.6B and 1.7B model nodes under the Cost-Sensitive mode, as well as bidirectional reward mode shifts (Q $\leftrightarrow$ C). Compared to the k-NN predictive router (the strongest baseline in the global UIS space), UNISCALE consistently maintains significantly lower stage-wise and final cumulative regret (see Table 6). Specifically, in the model removal environment (Figure 7), although the system loses its previously preferred low-cost nodes, UNISCALE rapidly explores and identifies new optimal configurations, achieving a 75.1% reduction in inference cost compared to the baseline while maintaining superior accuracy (+2.50pp). In the model additional and reward shift environments, UNISCALE exhibits exceptional agility by automatically triggering re-exploration mechanisms to fit new distributions. Further details on the dynamic evolution of UNISCALE are available in Section C.2, which provides a granular analysis of the policy recalibration process and recovery slopes following environmental drifts. This robustness confirms that the combination of semantic mapping and principled exploration allows the framework to consistently maintain an optimal quality–cost trade-off even under severe environmental perturbations.

For further comprehensive evaluations, please refer to Section C.4 for sensitivity to the verifier and Section C.5 for the physical latency breakdown.

## 6. Related Work and Discussion

**LLM Routing.** Early methods such as FrugalGPT (Chen et al., 2024a) and AutoMix (Aggarwal et al., 2024) relied

*Table 6.* Robustness comparison between UNISCALE and k-NN under non-stationary drifts involving action space dynamics (Add./Rem.) and reward mode shifts (Q ↔ C).

| Env. | Method | Reg.@130 | Reg.@210 | Acc. | Cost |
|---|---|---|---|---|---|
| Add. | UNISCALE | **27.43** | **38.52** | 43.75 | **59.6** |
| | k-NN | 34.42 | 51.73 | **45.62** | 805.1 |
| Rem. | UNISCALE | **27.56** | **41.15** | **53.12** | **255.9** |
| | k-NN | 35.52 | 57.49 | 50.62 | 1027.9 |
| Q → C | UNISCALE | **27.59** | **38.86** | **51.25** | **347.7** |
| | k-NN | 27.98 | 46.91 | 43.13 | 874.3 |
| C → Q | UNISCALE | **24.71** | **36.77** | **50.62** | **220.8** |
| | k-NN | 30.15 | 48.11 | 43.75 | 452.4 |

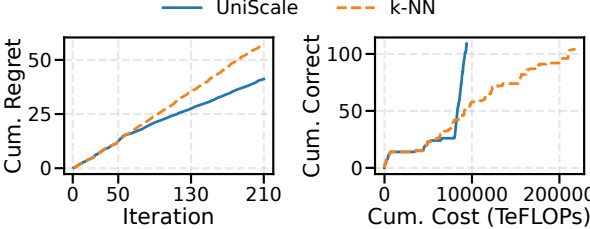

*Figure 7.* Cumulative regret and efficiency curves for UNISCALE and k-NN under a model removal environment. The 0.6B and 1.7B models are removed from the candidate set at the 51st iteration.

on cost-ordered cascading, sequentially querying models until a quality threshold was met. More recent work shifted toward data-driven routing, training lightweight predictors to assign queries to appropriate models, as exemplified by HybridLLM (Ding et al., 2024) and RouteLLM (Ong et al., 2025). Subsequent extensions improved the routing performance via contrastive learning or structured representations, e.g., RouterDC (Chen et al., 2024b) and GraphRouter (Feng et al., 2025). While BEST-Route (Ding et al., 2025) takes an initial step toward incorporating test-time scaling by using Best-of-N sampling, existing routers largely remain model-centric and rely on offline training.

**Test-Time Scaling.** TTS is generally categorized into two primary implementation pathways: serial scaling and parallel scaling. Serial scaling is characterized by the extension of reasoning chains (Wei et al., 2022), a strategy effectively employed by recent state-of-the-art models such as DeepSeek-R1 (Guo et al., 2025), OpenAI-o1 (Jaech et al., 2024), and QwQ (Team, 2025) to bolster complex reasoning capabilities. In contrast, parallel scaling encompasses techniques like repeated sampling and self-consistency (Wang et al., 2023; Brown et al., 2024), as well as reward-guided methods including Best-of-N, weighted voting, and tree search (Wan et al., 2024; Snell et al., 2025; Wu et al., 2025). While these approaches showcase the paradigm-shifting potential of TTS, current research remains largely confined to exploring performance ceilings of fixed models under static

budgets (Snell et al., 2025; Liu et al., 2025). The challenge of dynamically selecting optimal TTS strategies tailored to varying queries remains an open problem. UNISCALE is orthogonal to these techniques and can be combined with them. A detailed discussion is deferred to Section A.

## 7. Conclusion

We propose the *Unified Inference Scaling (UIS)* paradigm, which integrates model routing and Test-Time Scaling (TTS) into a unified decision space. To solve this complex online optimization problem, we design UNISCALE, *an adaptive framework based on the contextual multi-armed bandit*. By introducing efficiency-aware learning and cost modeling mechanisms, UNISCALE exploits the synergies between model routing and TTS to optimize the quality–cost frontier. Extensive experimental results demonstrate that UNISCALE *achieves fine-grained trade-offs across a broad spectrum in dynamic environments*. While the performance is currently modulated by the precision of Process Reward Models (PRMs), future work will explore more generalizable verification mechanisms to further enhance the framework's effectiveness across a broader range of tasks.

## Acknowledgements

We thank the anonymous reviewers for their insightful comments that helped improve this work. We also thank Zhaoyu Fan, Bowen Han, Zehua He, and Runze Lu for their helpful comments. This work was supported in part by the National Key Research and Development Program of China (Grant No. 2022YFA1003900), the Joint Funds of the National Natural Science Foundation of China (Grant No. U25A20394), the Science and Technology Commission of Shanghai Municipality (Grant Nos. 24DP1500704 and 24YL1901100), the "Medical+X" Interdisciplinary Research Project of Tongji University (Grant No. 2025-0674-YB-02), the Fundamental Research Funds for the Central Universities (Grant No. 22120230311), CUHK-Shenzhen Research Grant (Grant No. UDF01003466), the Guangdong Provincial Key Laboratory of Mathematical Foundations for Artificial Intelligence (Grant No. 2023B1212010001), the National Natural Science Foundation of China (Grant No. 62506319), the Guangdong Basic and Applied Basic Research Foundation (Grant No. 2026A1515030032), the Shenzhen Science and Technology Program (Grant No. JCYJ20250604141031003) , and the Pearl River Talent Program of Guangdong Province (Grant No. 2024QN11X069).

## Impact Statement

The profound significance of this work lies in proposing the *Unified Inference Scaling (UIS)* paradigm to break down technical silos between model routing and TTS, providing a

unified theoretical foundation and a *unified decision space* for large-scale AI inference orchestration. UNISCALE, as a concrete implementation mechanism, shifts inference optimization from heuristic-based rules to data-driven automated policies, laying the groundwork for building intelligent and standardized AI infrastructure.

**Environmental Sustainability (Green AI).** UNISCALE directly addresses the energy consumption challenges in the deployment of LLMs. By leveraging the *UIS Cost Model* to provide a unified equivalent FLOPs metric, the framework enables hardware-aware optimization across diverse execution environments. Furthermore, the *Path-Aware Early Exiting* mechanism minimizes redundant computation by dynamically terminating low-potential inference branches. This ensures that computational resources are concentrated on high-marginal-gain configurations, significantly reducing the carbon footprint of global inference services by preventing over-computation on suboptimal base models.

**Technological Democratization and Inclusivity.** This framework fosters a more inclusive AI ecosystem by providing a universal orchestration mechanism applicable to heterogeneous environments. By offering *fine-grained control* over the quality–cost trade-off, UNISCALE demonstrates significant potential in edge-cloud collaboration scenarios. Its *online adaptivity* allows resource-constrained edge hardware to intelligently determine when to leverage local capabilities and when to pursue cloud upgrades. This bridges the digital divide, enabling personal devices to perform complex reasoning tasks previously reserved for high-end clusters, thus allowing a broader user base to access advanced AI intelligence with lower barriers to entry.

**Data Privacy and Decentralized Governance.** While this research focuses on online learning, the standardized decision space abstracted by UNISCALE is inherently compatible with the federated orchestration paradigm (Dai et al., 2023; Huang et al., 2026). Since the Transformer-based encoder only processes abstract semantic features to determine configuration parameters without requiring access to raw corpora, it creates the possibility for building privacy-preserving cross-organizational inference networks. Sensitive query contexts and inference paths remain on local nodes, allowing global policy evolution to occur solely through the exchange of anonymized parameter updates.

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

r.press/v119/zhou20a.html.

# A. Additional Related Work

**Efficient Test-Time Scaling Strategies.** In the realm of efficient TTS strategies, researchers focus on constructing optimized reasoning spaces to enhance search efficiency. Monte Carlo Tree Search (MCTS) models the reasoning process as a heuristic search within a state space, leveraging reward models to guide the model beyond the limitations of traditional greedy decoding (Hao et al., 2023). AB-MCTS (Misaki et al., 2025) further introduces an adaptive mechanism that dynamically adjusts the search width and depth based on node uncertainty, achieving superior efficiency compared to standard MCTS. Forest of Thoughts (FoT) (Bi et al., 2025) designs a multi-tree reasoning framework based on sparse activation. By maintaining reasoning trees in parallel and expanding only the most relevant paths, it significantly reduces computational overhead while maintaining search breadth. From a theoretical perspective, Rebase (Wu et al., 2025) derives an optimal allocation strategy for TTS and proposes a reward-balanced tree search algorithm, proving that it outperforms traditional majority voting under a fixed FLOPs budget. However, existing works often remain confined to local optimizations of specific algorithms or fixed configurations, lacking a universal optimization of TTS. Bridging this gap, we adopt a unified perspective of the TTS inference process and propose UIS. This framework integrates model routing and TTS, facilitating a paradigm shift from algorithm-specific tuning to global parameterized search.

**Adaptive Resource Allocation in Test-Time Scaling.** Research in adaptive resource allocation aims to balance reasoning latency and computational cost through dynamic mechanisms. Adaptive-Consistency (Aggarwal et al., 2023) introduces a dynamic stopping mechanism based on statistical confidence (e.g., Beta distribution approximation), allowing the model to terminate sampling early once a consensus is reached, thereby drastically reducing redundant computation. IBPO (Yu et al., 2025) models the reasoning process as a utility maximization problem under budget constraints, enabling the model to perceive task difficulty and adaptively allocate reasoning length. This approach achieves significantly higher efficiency in solving complex mathematical problems compared to standard self-consistency methods. To further accelerate the generation process for complex tasks, SpecReason (Pan et al., 2025) applies speculative decoding at the reasoning-step level, utilizing a small model to quickly generate Chain-of-Thought (CoT) drafts which are then verified in parallel by a large model. R2R (Fu et al., 2025) proposes a neural token routing method that invokes the LLM only on identified divergent tokens along the critical path. Through dynamic collaboration with a Small Language Model (SLM), it achieves reasoning performance and speed that surpasses medium-sized models and approaches large-scale models with minimal active parameters. Despite these advances, most current mechanisms rely on offline training or static heuristic rules, making it difficult to respond in real-time to the environmental drift. UNISCALE addresses this via an online contextual bandit framework, achieving real-time joint optimization of model routing and TTS configurations.

# B. Additional Main Experimental Details and Results

## B.1. Infrastructure and Computational Environment

**Execution Framework.** All TTS strategies in this study are implemented within OpenR (Wang, 2024; Wang et al., 2024; Liu et al., 2025), an open-source framework specifically engineered for LLM reasoning. To facilitate efficient inference tasks, we optimized the model deployment engine by integrating the vLLM (v0.11.2) inference backend (Kwon et al., 2023). This engine implements highly efficient prefix caching and prefix sharing mechanisms and supports dynamic batching, significantly enhancing the computational efficiency of TTS strategies. All model weights and KV caches are loaded in BFloat16 (BF16) format to strike an optimal balance between numerical stability and memory efficiency.

**Hardware Platform.** Experimental evaluations were conducted on an NVIDIA A800 80GB SXM GPU cluster. To ensure hardware-referenced cost assessments, we utilize the eFLOPs (equivalent Floating Point Operations) cost model (Section D.3). We calibrated the base computational units according to the ratio of peak FP16/BF16 throughput to memory bandwidth of the NVIDIA A800 80GB SXM GPU (i.e., arithmetic intensity $I = 156$), ensuring that eFLOPs accurately reflect the hardware-level resource consumption across different model scales and TTS strategies.

## B.2. Semantic Representation of the Unified Inference Scaling Space

To enable effective reasoning across heterogeneous model capabilities and TTS strategies, we propose a semanticization pipeline that maps UIS space into a continuous manifold. Each UIS configuration $a = (M, QP, CP, BS)$ is first transformed into a structured textual description that captures the functional essence of both the backbone model $M$ and the TTS strategy $(QP, CP, BS)$. This description is then projected into a vector space.

**Model Specification.** For the model component $M$, we construct a capability-oriented description that includes both

architectural scale and empirical performance anchors, utilizing a key–value format that contains parameter scale (`Params`) and benchmark scores grouped by task categories such as expert reasoning (Rein et al., 2024; White et al., 2025), mathematics (Zhang & Math-AI, 2024; 2025), logic (Lin et al., 2025), and coding (Jain et al., 2025). To improve semantic alignment in the embedding space, each benchmark score is annotated with an explicit task-domain prefix, such as `ExpertReasoning_GPQA` or `Math_AIME24`. These task-level anchors serve as semantic references that allow the embedding model to associate a backbone LLM with its relative strengths across different reasoning dimensions, rather than treating the model identifier as an opaque symbol.

**Example (model description):**
```
Model: qwen3-14b | Params:  14B | ExpertReasoning_GPQA: 54.80 |
GeneralMixed_LiveBench:  59.60 | Math_AIME24:  31.70 | Logic_Zebra:  33.00
| Coding_LCB: 29.00
```

**TTS Configuration.** The TTS component $(QP, CP, BS)$ is translated into a structured description that emphasizes its functional semantics rather than its raw numerical values. We reinterpret $QP$ as the number of `Parallel_Trees`, $CP$ as the number of `Path_Candidates`, and $BS$ as the effective `Beam_Width` controlling pruning granularity. In addition, we introduce derived attributes that make implicit interactions explicit. For instance, the effective resource amplification factor is computed as $\min(QP \cdot CP, 64)$, and the number of expansions per step is derived from $CP/BS$. Based on these quantities, each configuration is assigned a high-level `Strategy_Mode` (e.g., `Fast-Inference`, `Balanced-Search`, `Deep-Reasoning`) and an associated `Optimization_Priority` (e.g., `Latency-First`, `Balanced-Efficiency`, `Accuracy-First`).

**Example (TTS description):**
```
Parallel_Trees(QP): 4 | Path_Candidates(CP): 16 | Beam_Width(BS): 4
| Expansions_per_Step: 4 | Resource_Impact:  16x | Strategy_Mode:
Balanced-Search | Optimization_Priority:  Balanced-Efficiency
```

**Action Semantic Representation.** We concatenate the model and TTS descriptions to form the final semantic description, which is then mapped into a 1024-dimensional semantic vector for each action using a pretrained text embedding model (`qwen-text-embedding-v4`). This embedding process transforms the discrete, high-dimensional configuration space into a continuous latent space where semantic proximity reflects functional similarity. Unlike direct numerical encodings, this approach enables cross-model generalization by embedding diverse backbone models into a unified representation based on their empirical capability profiles. It also facilitates strategy-level similarity awareness, ensuring that TTS configurations with analogous resource allocation patterns are positioned closely in the vector space, regardless of their specific parameter values. Furthermore, the 1024-dimensional action embedding serves as a rich input for downstream bandit or regression models, allowing them to effectively model the complex non-linear interactions between model strength and inference-time scaling strategies. Consequently, the resulting vector provides a coherent and information-dense representation that supports efficient reward estimation and exploration within the UNISCALE framework.

### B.3. Hyperparameters and Baseline Configurations

**Hyperparameter Details for UNISCALE.** We established the following key hyperparameters:

1. **Exploration Factor** ($\alpha$): Set to 1.0 to ensure sufficient exploration of the action space during the initial stages while facilitating rapid convergence to the optimal policy in later stages.

2. **Regularization Term** ($\lambda$): The regularization coefficient for ridge regression is set to 1.0 to maintain numerical stability during the Gram matrix $\mathbf{A}$ inversion.

**Reward Function Hyperparameters.** We defined two typical reward modes:

1. **Cost-Sensitive**: $w_1 = w_2 = 0.1, w_3 = 0.8$. This mode aims to identify the most cost-effective UIS configuration.

2. **Cost-Leaning**: $w_1 = w_2 = 0.2, w_3 = 0.6$. This mode is designed to favor cost efficiency while preserving acceptable performance.

3. **Quality-Leaning**: $w_1 = w_2 = 0.3, w_3 = 0.4$. This mode aims to emphasize inference quality while incorporating moderate cost considerations.

4. **Quality-Priority**: $w_1 = w_2 = 0.4, w_3 = 0.2$. This mode is designed to identify the UIS configurations with the highest inference quality.

**Baseline Configurations.** Following the predictive routing paradigms in RouterBench (Hu et al., 2024a), we configured the predictive routing baselines as follows:

1. **MLP**: The MLP has an input dimension of 2048, an output dimension of 1, and a hidden layer of size 100. We train the MLP to minimize the mean squared error (MSE) loss for 1000 iterations after each new observation point $(\mathbf{x}_{t,a_t}, r_t)$. A default learning rate of 0.001 is used.

2. **k-NN**: We implement a k-NN router with $k = 5$, which estimates the reward by averaging the outcomes of the $k$ most similar historical instances in the joint feature space:

$$a_t = \arg\max_{a \in \mathcal{A}} \left( \frac{1}{k} \sum_{i \in \mathcal{N}(\mathbf{x}_{t,a})} r_i \right), \tag{5}$$

where $\mathcal{N}(\mathbf{x}_{t,a})$ denotes the set of $k$ indices $i < t$ whose historical vectors $\mathbf{x}_{i,a_i}$ exhibit the highest cosine similarity to the current candidate vector $\mathbf{x}_{t,a}$.

3. **Thompson Sampling** (Chapelle & Li, 2011): We implement a linear Thompson Sampling strategy with Gaussian posterior sampling. At each iteration, the algorithm samples a parameter vector $\tilde{\boldsymbol{\theta}}_t$ from the posterior distribution:

$$\tilde{\boldsymbol{\theta}}_t \sim \mathcal{N}(\hat{\boldsymbol{\theta}}_t, \alpha^2 \mathbf{A}_t^{-1}), \tag{6}$$

where $\hat{\boldsymbol{\theta}}_t = \mathbf{A}_t^{-1} \mathbf{b}_t$ denotes the posterior mean estimated from historical observations. The action is then selected according to:

$$a_t = \arg\max_{a \in \mathcal{A}} \mathbf{x}_{t,a}^\top \tilde{\boldsymbol{\theta}}_t. \tag{7}$$

After observing the reward $r_t$, the posterior statistics are updated online using the newly observed tuple $(\mathbf{x}_{t,a_t}, r_t)$. We use the same regularization term $\lambda$ as UNISCALE, with $\lambda = 1.0$, and set the exploration factor $\alpha$ to 1.0.

4. **NeuralUCB** (Zhou et al., 2020): We implement NeuralUCB based on the same MLP architecture and training configuration as the MLP baseline. At each iteration, the action is selected according to the upper confidence bound:

$$a_t = \arg\max_{a \in \mathcal{A}} \left( f(\mathbf{x}_{t,a}; \boldsymbol{\theta}_t) + \alpha \sqrt{\mathbf{g}(\mathbf{x}_{t,a})^\top \mathbf{Z}_t^{-1} \mathbf{g}(\mathbf{x}_{t,a})} \right), \tag{8}$$

where $f(\mathbf{x}_{t,a}; \boldsymbol{\theta}_t)$ denotes the neural reward predictor, and $\mathbf{g}(\mathbf{x}_{t,a}) = \nabla_{\boldsymbol{\theta}} f(\mathbf{x}_{t,a}; \boldsymbol{\theta}_t)$ represents the gradient feature used for uncertainty estimation. After each interaction step, the model is updated online using all historical observations. We set the exploration factor $\alpha$ to 1.0.

### B.4. Detailed Results on Main Experiment

The dynamic curves in Figure 8 reveal distinct behavioral patterns for each baseline across different scaling scenarios and reward modes.

**Greedy Baseline.** This approach excels in low-dimensional and sparse action spaces, such as Model Routing. Its performance closely matches UNISCALE in these settings, as the reward estimator accurately identifies optimal models during the warm-up phase without requiring further exploration. However, in high-dimensional spaces (TTS and UIS), Greedy suffers from a significant exploration gap, resulting in higher cumulative regret as it fails to navigate complex configuration boundaries.

**MLP Baseline.** MLP shows competitive performance only in Adaptive TTS, where the reward landscape is relatively smooth and monotonic with respect to the scale of inference structures. In the more complex scenarios, it exhibits high

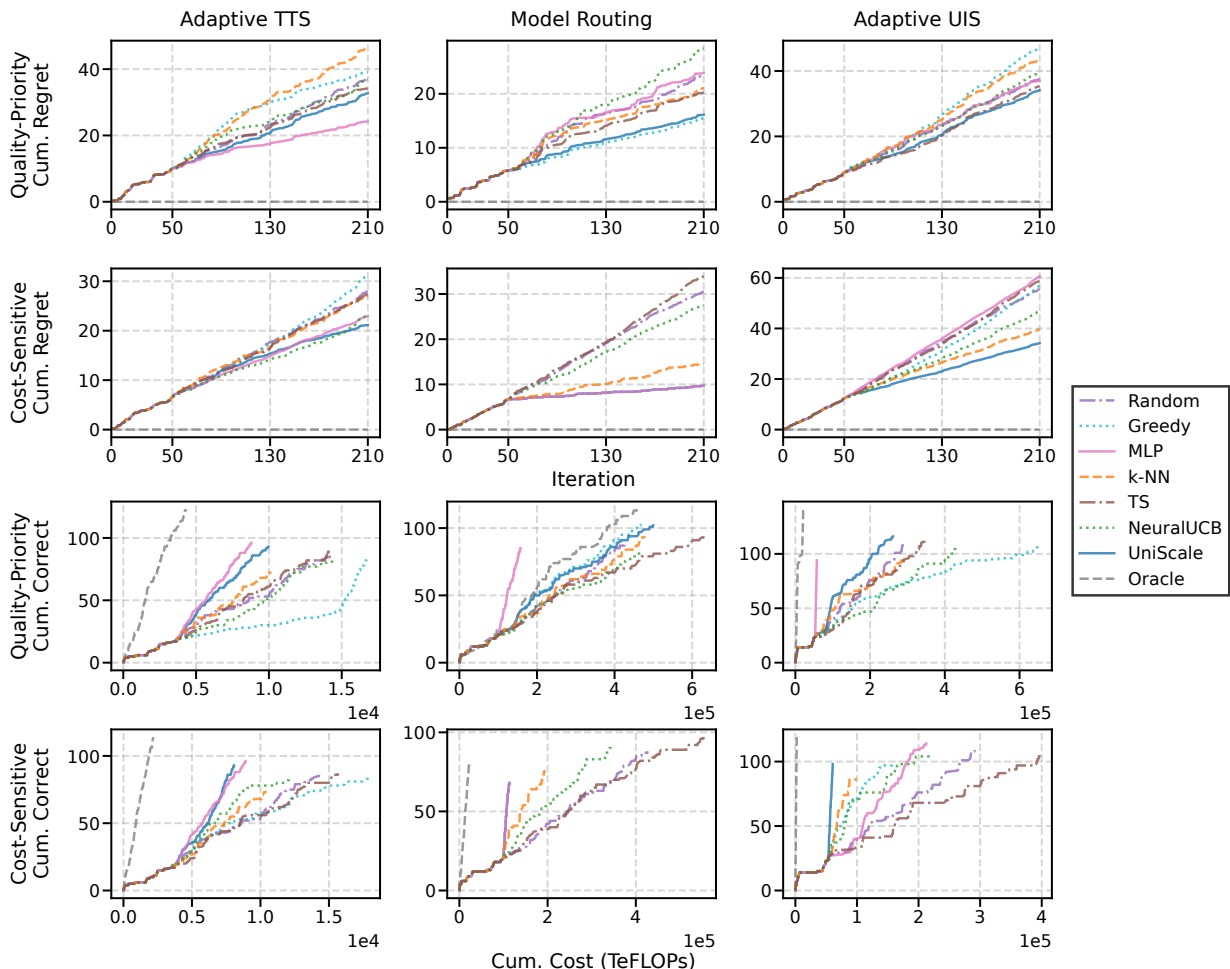

*Figure 8.* Performance comparison of UNISCALE and baselines across various scaling paradigms. The columns represent Adaptive TTS, Model Routing, and Adaptive UIS, respectively. The top two rows display the cumulative regret versus iterations, while the bottom two rows illustrate the cost-benefit efficiency (Cumulative Correct counts versus Cumulative Cost in TeFLOPs). Results are evaluated under both Quality-Priority and Cost-Sensitive reward modes.

variance and instability. The discrete performance jumps between heterogeneous model architectures lead to prolonged performance plateaus and erratic jumps in efficiency, highlighting the difficulty of fitting non-continuous joint spaces with sparse online samples.

**k-NN Baseline.** k-NN is highly effective in Cost-Sensitive mode across all scenarios. This is because inference costs are naturally clustered in the semantic space, allowing passive similarity matching to efficiently locate low-cost configurations. Conversely, it underperforms in Quality-Priority mode because the lack of an active exploration mechanism like UNISCALE prevents it from proactively discovering non-linear, query-specific accuracy peaks, leaving it limited by the distribution of historical samples.

### B.5. Additional Comparisons with BEST-Route

To comprehensively evaluate the advantages of the UIS paradigm, we conduct an additional comparison against BEST-Route. To ensure a compatible evaluation, we define a BEST-Route* scenario that adopts a restricted search space consistent with the original BEST-Route framework. Specifically, the router is constrained to select a single configuration from 120 candidates (6 models × 20 TTS actions). These include six Qwen3 models (0.6B, 1.7B, 4B, 8B, 14B, 32B) and a Best-of-N strategy with $N \in \{1, \ldots, 20\}$, parameterized as $QP \in \{1, \ldots, 20\}$ with $CP = BS = 1$. Furthermore, Skywork-o1-Open-PRM-Qwen-2.5-1.5B is employed as the underlying verifier for process evaluation.

*Table 7.* Main performance comparison across BEST-Route* and UIS scenarios. Results are reported under Cost-Sensitive and Quality-Priority reward modes (excluding the 50-step warm-up phase).

| Method | Metric | Cost-Sensitive | | Quality-Priority | |
|---|---|---|---|---|---|
| | | BEST-Route* | UIS | BEST-Route* | UIS |
| UNISCALE (ours) | Reward (↑) | 0.5500 | 0.6973 | 0.5507 | 0.6326 |
| | Accuracy (↑) | 35.00 | 46.25 | 50.63 | 57.50 |
| | Cost (↓) | 1119.8 | 48.2 | 3537.7 | 1303.3 |
| Oracle | Reward (↑) | 0.6279 | 0.8335 | 0.6504 | 0.7899 |
| | Accuracy (↑) | 42.50 | 56.87 | 59.38 | 67.50 |
| | Cost (↓) | 103.6 | 10.4 | 3163.9 | 113.9 |

*Table 8.* Main performance comparison on the coding task under the Cost-Sensitive reward mode within the UIS scenario.

| Method | Reward (↑) | Accuracy (↑) | Cost (↓) |
|---|---|---|---|
| k-NN | 0.7943 | **53.13** | 14039.5 |
| UNISCALE | **0.8125** | 52.50 | **5619.6** |
| *Relative Change* | +0.0182 | -0.63pp | -59.97% |

As demonstrated by the theoretical Oracle performance, BEST-Route's space has a significantly lower upper bound than the global UIS space. Under the Cost-Sensitive mode, the full UIS space achieves a notable accuracy improvement (+14.37pp) while consuming only roughly 10% of the computational cost compared to the BEST-Route* space. Similarly, in the Quality-Priority mode, the UIS space elevates the accuracy ceiling (+8.12pp) at only 3.6% of the corresponding cost. This confirms that relying solely on one-dimensional test-time scaling restricts the system's capacity to optimize the fine-grained quality–cost frontier.

Notably, UNISCALE also performs effectively when exploring within the BEST-Route* space, efficiently identifying the best available configurations given the spatial constraints. However, the framework's full potential is uniquely unlocked when operating in the joint UIS space. In the Cost-Sensitive mode, UNISCALE operating in the full UIS space achieves a double-digit accuracy gain (+11.25pp) while eliminating over 95% of the inference overhead compared to its performance in the restricted space. In the Quality-Priority mode, the unified approach yields an additional accuracy boost (+6.87pp) while simultaneously reducing the computational cost by more than 60%. These empirical results validate that our joint optimization fundamentally expands the efficiency boundaries beyond earlier routing paradigms.

### B.6. Generalization Across Diverse Tasks

To validate the task-agnostic generalizability of UNISCALE, we conduct an empirical evaluation on a coding benchmark as a representative case study. For this setup, the candidate model pool is selected as a subset of those in Table 1, specifically comprising Qwen3-4B and Qwen3-8B, while the available TTS strategies and the verifier remain identical to our primary experiments. The evaluation dataset comprises 210 instances sampled from LiveCodeBench (Jain et al., 2025). Following the official evaluation paradigm, it consists of 77 standard-input and 133 call-based instances, classified into 91 easy and 119 medium problems. These 210 instances directly correspond to the experimental workflow, which consists of a 50-step warm-up followed by 160 policy-driven iterations. All trials are evaluated under the Cost-Sensitive reward mode within the full joint UIS optimization space.

The empirical comparisons are reported in Table 8. The results clearly highlight UNISCALE's exceptional data efficiency and orchestration capability under a completely different task profile. Compared to the competitive k-NN baseline, UNISCALE yields a remarkable improvement in operational efficiency, slashing the total computational cost by -59.97%. Although this stringent cost restriction leads to a marginal degradation in absolute accuracy (-0.63pp), the joint optimization mechanism effectively balances the quality–cost trade-off, culminating in an overall gain of +0.0182 in mean reward. This successful alignment under a non-mathematical task configuration firmly substantiates that UNISCALE's core scheduling logic is inherently task-agnostic. We leave the extensive evaluation and methodological scaling toward broader open-ended task environments to future work.

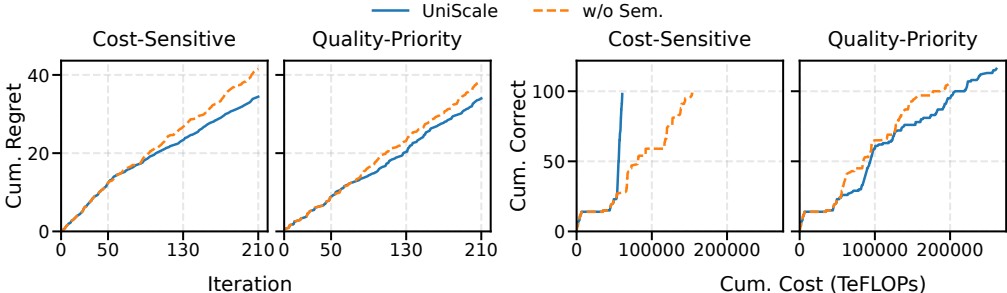

*Figure 9.* Performance comparison between UNISCALE and a non-semantic baseline (*w/o Sem.*) across different reward modes. The left panels display cumulative regret versus iterations, while the right panels illustrate cumulative correct counts versus cumulative inference cost (TeFLOPs).

## C. Detailed Results on Ablation Study

### C.1. Detailed Results for Effectiveness of Action Semantic Representations

Figure 9 provides the comprehensive performance visualizations for action semantic representations across both Cost-Sensitive and Quality-Priority modes.

**Learning Trajectory (Left Panels).** The cumulative regret curves illustrate that the advantage of action semantics emerges immediately following the warm-up phase. The consistently lower slope of UNISCALE compared to the *w/o Sem.* baseline confirms that mapping configurations into a unified semantic space allows for efficient cross-action knowledge transfer, reducing the exploration overhead in high-dimensional action spaces.

**Marginal Cost-Benefit (Right Panels).** The efficiency curves visualize the trade-off range discussed in Section 5. Notably, in Cost-Sensitive mode, UNISCALE exhibits a near-vertical ascent in cumulative correctness at extremely low cost levels. In Quality-Priority mode, the curves demonstrate that UNISCALE achieves a higher performance ceiling than the baseline, proving that semantic awareness allows the system to identify high-quality configurations that are otherwise difficult to locate via independent one-hot encoding.

### C.2. Detailed Results for Robustness to Non-stationary Environmental Drifts

Figure 10 illustrates the dynamic performance evolution of UNISCALE and k-NN across four typical non-stationary environments. The online adaptability of the framework is intuitively reflected through the changes in the slopes of the cumulative regret curves (top row) and efficiency curves (bottom row). During the warm-up phase (iterations 1–50), the cumulative regret slopes are nearly identical and rise consistently across all subplots, reflecting the parity in environmental perception when both algorithms employ the same random exploration strategy. Upon reaching the environmental drift point at the 51st iteration, their decision trajectories diverge sharply:

**Action Space Dynamics.** In the Model Additional environment (under Cost-Sensitive mode), UNISCALE rapidly identifies and exploits the newly introduced low-cost Qwen-0.6B and 1.7B nodes. Consequently, its cumulative regret slope flattens significantly immediately after the drift, whereas the reduction in the slope of k-NN's regret growth lags behind, indicating weaker adaptability to action space expansion. In the Model Removal environment, the loss of low-cost nodes causes a sudden spike in the regret slopes for both methods. However, UNISCALE leverages policy recalibration within the semantic space to quickly bring its slope back down to a level lower than that of the warm-up phase. In contrast, the regret slope for k-NN remains consistently higher than its initial warm-up rate.

**Reward Mode Shifts.** When transitioning from Quality-Priority to Cost-Sensitive, UNISCALE initially maintains a regret slope slightly higher than that of k-NN due to the activation of its re-exploration mechanism. However, it eventually converges to a much flatter slope than the baseline, allowing its cumulative regret to overtake k-NN before the 130th iteration. In the transition from Cost-Sensitive to Quality-Priority, UNISCALE similarly demonstrates agile adaptability. Its regret slope stabilizes rapidly after a brief fluctuation, proving the algorithm's ability to precisely locate and smoothly migrate to high-performance configuration regions, thereby effectively compensating for the performance loss caused by the mode

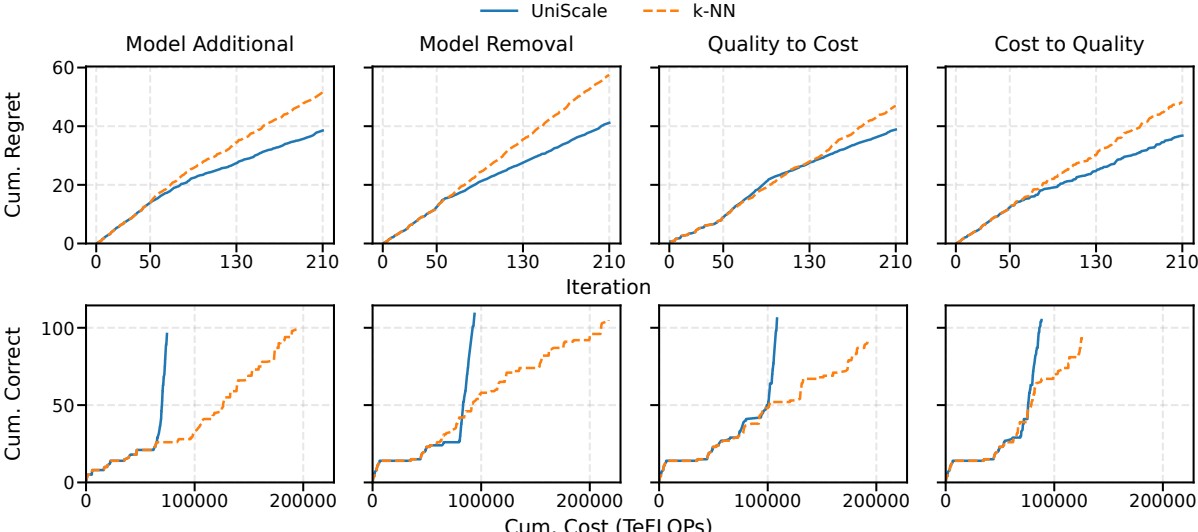

*Figure 10.* Robustness comparison between UNISCALE and k-NN under non-stationary drifts. The top row displays cumulative regret versus iterations, and the bottom row shows cumulative correct counts versus inference cost (TeFLOPs). Environmental drifts are introduced at the 51st iteration, encompassing action space dynamics (Model Addition/Removal) and reward mode shifts (between Quality-Priority and Cost-Sensitive).

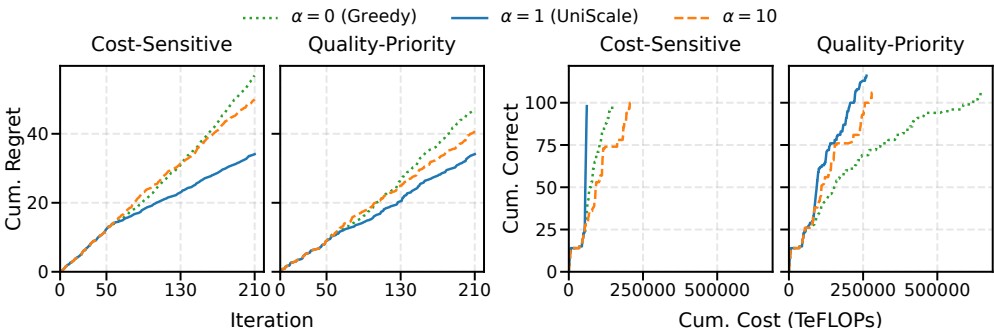

*Figure 11.* Sensitivity analysis of the exploration factor $\alpha$. The left panels show the cumulative regret over iterations, while the right panels illustrate the cumulative correct counts versus cumulative inference cost (TeFLOPs). Results are compared across Cost-Sensitive and Quality-Priority modes. $\alpha = 1$ represents the default configuration of UNISCALE, while $\alpha = 0$ corresponds to a purely Greedy strategy.

switch.

Collectively, these results demonstrate that UNISCALE can consistently select superior UIS configurations through wide-range, fine-grained quality–cost trade-offs in dynamic and non-stationary production environments, thereby maximizing the overall efficacy of the UIS paradigm.

### C.3. Sensitivity analysis of the exploration factor $\alpha$

Figure 11 illustrates the impact of the exploration factor $\alpha$ on algorithm performance. $\alpha$ governs the trade-off between exploiting known high-reward configurations and exploring uncertain regions of the action space.

$\alpha = 0$ **(Purely Greedy).** In the left panel, the cumulative regret curves for $\alpha = 0$ (dotted line) exhibit a higher growth rate following the warm-up phase. This suggests that without an active exploration mechanism, the system becomes trapped in local optima, failing to discover higher-reward actions within the UIS space and resulting in suboptimal efficiency boundaries

*Table 9.* Main performance comparison across different PRMs under Cost-Sensitive and Quality-Priority reward modes (excluding the 50-step warm-up phase). Specifically, PRM-1.5B and PRM-7B denote configurations using Skywork-o1-Open-PRM-Qwen-2.5-1.5B and Skywork-o1-Open-PRM-Qwen-2.5-7B as the verifiers, respectively. **Best** results are highlighted.

| Method | Metric | Cost-Sensitive | | Quality-Priority | |
|---|---|---|---|---|---|
| | | UIS (PRM-7B) | UIS (PRM-1.5B) | UIS (PRM-7B) | UIS (PRM-1.5B) |
| k-NN | Reward ($\uparrow$) | 0.6615 | 0.6542 | 0.6128 | 0.5740 |
| | Accuracy ($\uparrow$) | 41.25 | 39.37 | 55.63 | 43.75 |
| | Cost ($\downarrow$) | 249.8 | 148.7 | 1235.3 | 996.2 |
| UNISCALE (ours) | Reward ($\uparrow$) | **0.7092** | **0.6973** | **0.6562** | **0.6326** |
| | Accuracy ($\uparrow$) | 48.13 | 46.25 | **63.75** | **57.50** |
| | Cost ($\downarrow$) | **79.9** | **48.2** | 1485.8 | 1303.3 |

*Table 10.* Latency breakdown of UNISCALE components in the UIS space (Quality-Priority mode). Results show the cumulative execution time across 160 post-warm-up iterations.

| Metric | Query Embedding | Bandit Overhead | LLM Inference | PRM Verification | TTS Overheads | Total Time |
|---|---|---|---|---|---|---|
| Time (s) | 16.79 | 3.15 | 1941.84 | 175.31 | 49.23 | 2186.31 |
| Ratio | 0.77% | 0.14% | 88.82% | 8.02% | 2.25% | 100.00% |

in the right panel.

$\alpha = 10$ **(Excessive Exploration).** While covering more of the search space, the left panel shows that $\alpha = 10$ consistently maintains a high level of cumulative regret. The efficiency curves in the right panel reveal a significant exploration overhead, where the algorithm consumes excessive computational budget on suboptimal configurations, leading to a slower ascent in cumulative correctness compared to $\alpha = 1$.

$\alpha = 1$ **(Balanced Exploration).** Across all reward modes, $\alpha = 1$ (solid line) consistently achieves the lowest cumulative regret (left panel) and the steepest efficiency trajectory (right panel). These results confirm that a balanced, principled exploration intensity is essential for accelerating policy optimization and achieving superior wide-range quality–cost trade-offs.

### C.4. Sensitivity to the Underlying Verifier

To verify the robustness and compatibility of the framework within the adaptive UIS scenario, we evaluate UNISCALE against the k-NN baseline across different Process Reward Model (PRM-7B and PRM-1.5B), as detailed in Table 9. The empirical results demonstrate that UNISCALE consistently delivers superior trade-off efficiency while highlighting distinct behaviors across varying verifier capacities:

**Cost-Sensitive Mode.** Under this configuration, UNISCALE achieves a remarkable reduction in inference overhead while simultaneously improving accuracy over the baseline. Specifically, it cuts the computational cost significantly (-68%) for both the PRM-7B and PRM-1.5B verifiers. When comparing UNISCALE's performance across the two PRM scales in this mode (where LLM inference is lightweight), the additional cost of using the larger verifier is more pronounced (cost +66%, accuracy +1.88 pp). This highlights UNISCALE's capability to effectively steer policies toward low-cost configuration spaces under constrained budgets.

**Quality-Priority Mode.** When the system shifts toward prioritizing generation accuracy, UNISCALE effectively pushes the performance ceiling of the UIS paradigm. Compared to the k-NN baseline, it elevates the evaluation accuracy (+8.12 pp) for the PRM-7B verifier and yields a significant absolute improvement (+13.75 pp) for the PRM-1.5B verifier. Furthermore, scaling up the underlying PRM within UNISCALE in this mode (where LLM inference dominates) results in a modest cost increase (+14%) alongside a significant accuracy improvement (+6.25 pp). Although prioritizing quality inherently drives up the absolute test-time cost, UNISCALE consistently maintains a superior overall reward-cost trade-off compared to the baseline.

Overall, these results show that while stronger PRMs can further enhance performance, UNISCALE remains highly effective even with weaker verifiers, indicating limited sensitivity to PRM quality and robustness to verification noise.

## C.5. System Overhead and Latency Breakdown

To assess the practical deployment efficiency of UNISCALE, we analyze the cumulative latency across internal components. In Table 10, Query Embedding corresponds to the semantic mapping defined in Section 3.1. Bandit Overhead encompasses both the reward estimator update in Section 3.1 and the action acquisition step using the LinUCB algorithm in Section 3.2. Within the execution phase outlined in Section 3.3, LLM Inference maps to State Generation, PRM Verification corresponds to Process Verification, and TTS Overheads encapsulate the remaining operational steps.

Crucially, the total online overhead introduced by UNISCALE's core orchestration modules, specifically the combined latency of Query Embedding and Bandit Overhead, accounts for a mere 0.91% of the total execution time. This exceptional efficiency is primarily achieved through two mechanisms: first, action semantic embeddings are pre-computed offline and excluded from the runtime overhead, meaning that real-time processing only requires a query embedding process to form the final joint representation of dimension $d = 2048$; second, we utilize the Sherman-Morrison formula for rank-1 bandit updates, which requires only $\mathcal{O}(d^2)$ operations instead of an explicit $\mathcal{O}(d^3)$ matrix inversion. This confirms that the framework components introduced by UNISCALE impose a negligible computational footprint relative to the core generative process.

# D. More Details on UNISCALE Algorithm

## D.1. More Details on Principled Uncertainty Measure

This section provides a formal derivation of the principled uncertainty metric $\sqrt{\mathbf{x}_{t,a}^\top \mathbf{A}_t^{-1} \mathbf{x}_{t,a}}$ (also denoted as $\|\mathbf{x}_{t,a}\|_{\mathbf{A}_t^{-1}}$), alongside its physical interpretation within the UIS space.

**Mathematical Derivation.** To establish a rigorous foundation for the reward prediction mechanism in UNISCALE, we begin by characterizing the underlying reward generating process.

**Assumption D.1 (Linear Reward and sub-Gaussian Noise).** There exists an unknown true parameter vector $\boldsymbol{\theta}^* \in \mathbb{R}^d$ such that for any action $a$ with its feature vector $\mathbf{x}_{t,a}$, the observed reward $r_{t,a}$ satisfies a linear relationship:

$$r_{t,a} = \mathbf{x}_{t,a}^\top \boldsymbol{\theta}^* + \eta_t, \tag{9}$$

where $\eta_t$ represents a $\sigma$-sub-Gaussian random noise reflecting the stochastic nature of the environment.

Based on this linear assumption, we can utilize ridge regression to estimate the unknown vector $\boldsymbol{\theta}^*$ from historical observations.

**Definition D.2 (Ridge Regression Estimator).** Given the history of observations up to time $t - 1$, denoted as $\mathcal{H}_{t-1} = \{(\mathbf{x}_{\tau,a_\tau}, r_\tau)\}_{\tau=1}^{t-1}$, we define the Gram matrix $\mathbf{A}_t$ and the cumulative reward-weighted vector $\mathbf{b}_t$ as:

$$\mathbf{A}_t = \lambda \mathbf{I} + \sum_{\tau=1}^{t-1} \mathbf{x}_{\tau,a_\tau} \mathbf{x}_{\tau,a_\tau}^\top, \tag{10}$$

$$\mathbf{b}_t = \sum_{\tau=1}^{t-1} r_\tau \mathbf{x}_{\tau,a_\tau}, \tag{11}$$

where $\lambda > 0$ is the regularization parameter. The ridge regression estimate of the parameter vector $\boldsymbol{\theta}$, denoted as $\hat{\boldsymbol{\theta}}_t$, is given by:

$$\hat{\boldsymbol{\theta}}_t = \mathbf{A}_t^{-1} \mathbf{b}_t. \tag{12}$$

To quantify the uncertainty of this estimate, we introduce the following norm to measure distances in the feature-weighted space.

**Definition D.3 (Mahalanobis Norm).** For a positive definite matrix $\mathbf{A}$, the weighted norm of a vector $\mathbf{z}$ is defined as $\|\mathbf{z}\|_{\mathbf{A}} = \sqrt{\mathbf{z}^\top \mathbf{A} \mathbf{z}}$.

Building upon linear bandit theory (Abbasi-Yadkori et al., 2011), the relationship between our estimate $\hat{\boldsymbol{\theta}}_t$ and the true parameter $\boldsymbol{\theta}^*$ can be bounded within a high-probability region.

**Lemma D.4** (**Confidence Ellipsoid**). *Under Assumption D.1, for any $\delta \in (0, 1)$, the true parameter $\boldsymbol{\theta}^*$ resides within a confidence ellipsoid $\mathcal{E}_t$ centered at $\hat{\boldsymbol{\theta}}_t$ with probability at least $1 - \delta$:*

$$\mathcal{E}_t = \left\{ \boldsymbol{\theta} : \|\hat{\boldsymbol{\theta}}_t - \boldsymbol{\theta}\|_{\mathbf{A}_t} \leq \beta_t \right\}, \tag{13}$$

*where $\beta_t$ is a scaling factor that depends on the time step $t$ and the desired confidence level $\delta$.*

By projecting this ellipsoid onto the direction of a new action's feature vector, we obtain a formal bound for the reward prediction error.

**Proposition D.5** (**Reward Deviation Bound**). *For any candidate feature vector $\mathbf{x}_{t,a}$, the deviation between the predicted reward $\mathbf{x}_{t,a}^\top \hat{\boldsymbol{\theta}}_t$ and the expected true reward $\mathbf{x}_{t,a}^\top \boldsymbol{\theta}^*$ is bounded by the uncertainty of the action in the current semantic space:*

$$|\mathbf{x}_{t,a}^\top \hat{\boldsymbol{\theta}}_t - \mathbf{x}_{t,a}^\top \boldsymbol{\theta}^*| \leq \|\mathbf{x}_{t,a}\|_{\mathbf{A}_t^{-1}} \|\hat{\boldsymbol{\theta}}_t - \boldsymbol{\theta}^*\|_{\mathbf{A}_t} \leq \beta_t \sqrt{\mathbf{x}_{t,a}^\top \mathbf{A}_t^{-1} \mathbf{x}_{t,a}} \tag{14}$$

*Proof.* This follows from the generalized Cauchy-Schwarz inequality, $|\mathbf{u}^\top \mathbf{v}| \leq \|\mathbf{u}\|_{\mathbf{A}^{-1}} \|\mathbf{v}\|_{\mathbf{A}}$, by setting $\mathbf{u} = \mathbf{x}_{t,a}$ and $\mathbf{v} = \hat{\boldsymbol{\theta}}_t - \boldsymbol{\theta}^*$. $\square$

**Physical Interpretation.** This exploration term carries explicit semantic meaning within the UIS space:

1. **Data-Driven Exploration Decay**: The Gram matrix $\mathbf{A}_t$ encodes the density of historical samples in the semantic space. When the system frequently samples a specific UIS configuration, the eigenvalues of $\mathbf{A}_t$ along the corresponding feature dimensions increase.

2. **Semantic Knowledge Transfer**: Since UNISCALE employs a Transformer encoder to extract action semantic features $\mathbf{s}_a$, the exploration term $\|\mathbf{x}_{t,a}\|_{\mathbf{A}_t^{-1}}$ can recognize actions with structural similarities. For instance, even if a specific UIS configuration has never been selected, the system can automatically infer its uncertainty based on its proximity to previously explored actions in the semantic embedding space (e.g., similar model specifications or TTS strategies), thereby accelerating convergence.

3. **Robustness under Environmental Drift**: In the event of environmental drift, newly emerging feature vectors $\mathbf{s}_{q_t}$ will have low coverage in the historical data. This causes $\|\mathbf{x}_{t,a}\|_{\mathbf{A}_t^{-1}}$ to increase instantaneously, triggering a re-exploration mechanism that allows the system to rapidly adapt to the new environment.

### D.2. Verifier Scores as an Indicator of Correctness

Statistical analysis based on 105840 large-scale samples (encompassing 210 questions across 168 configurations with 3 sampling iterations each) reveals that verifier scores exhibit significant distributional discretizations across different correctness categories, as shown in Figure 12. Correct answers demonstrate a pronounced high-confidence clustering effect, characterized by a high mean score of 0.8818 and a narrow standard deviation of 0.1830, with samples highly skewed toward a narrow frequency band near 1.0. This reflects the verifier's consistent preference and stable confidence regarding correct logical paths. In contrast, the scoring distribution for incorrect answers is notably flatter and more dispersed, with the mean decreasing to 0.5961 and the standard deviation expanding to 0.2934, indicating substantially higher uncertainty and volatility. This systematic shift in distribution morphology suggests that when encountering incorrect answers, the verifier's scoring behavior is driven by uncertainty-induced random diffusion rather than systematic high-score misjudgment. These distributional phenomena are further corroborated by robust quantitative metrics: the point-biserial correlation coefficient between Verifier Score and answer correctness reaches 0.5061 ($P < 0.001$), indicating a significant and moderately strong statistical relationship, while the corresponding AUROC (Area Under the Receiver Operating Characteristic curve) of 0.8067 confirms the score's reliable discriminative capacity. Consequently, the verifier score provides a stable and exploitable signal for overall ranking and selection, effectively distinguishing correctness.

### D.3. Unified Inference Scaling Cost Model

To evaluate the resource consumption of the UNISCALE framework, we propose a cost model based on equivalent FLOPs (eFLOPs) (Sadhukhan et al., 2025). This section formalizes the derivation of the total inference cost $C_{\text{UIS}}$ through a series of definitions, assumptions, and propositions.

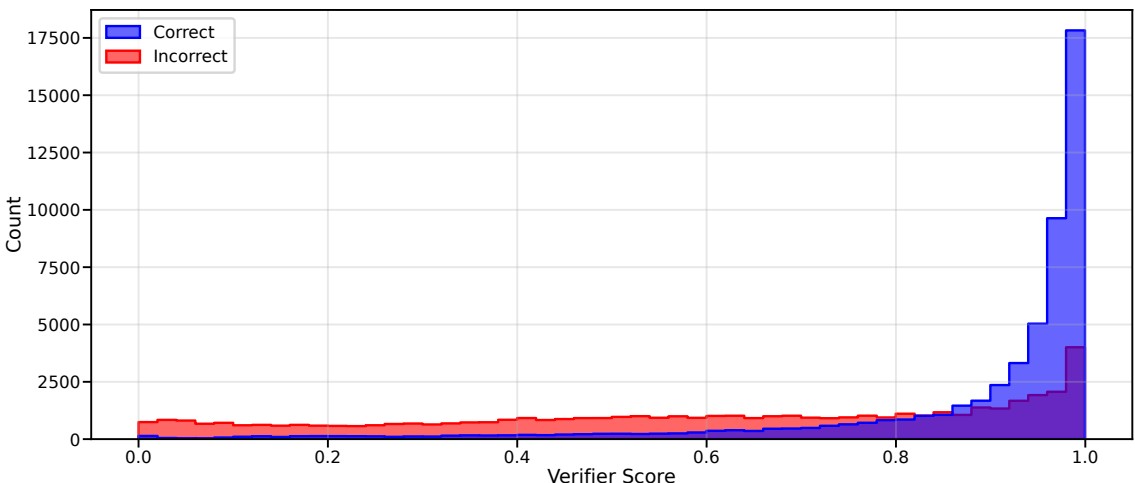

*Figure 12.* Distribution of verifier scores for correctness assessment. Correct answers (blue) cluster near 1.0 with higher confidence, while incorrect answers (red) show a flatter distribution.

**Definition D.6 (Equivalent FLOPs, eFLOPs).** To unify compute-bound and memory-bound overheads in LLM inference, we define eFLOPs as:

$$\text{eFLOPs} = C_{\text{comp}} + C_{\text{mem}} \cdot I, \tag{15}$$

where $C_{\text{comp}}$ denotes the number of floating-point operations, $C_{\text{mem}}$ represents the memory access volume in bytes, $I$ denotes the arithmetic intensity of the hardware (e.g., NVIDIA A800 80GB SXM GPU), which is defined as the ratio between peak FLOPS and memory bandwidth.

**Definition D.7 (Intermediate State).** An intermediate state during the inference process is denoted as $s_{i,j,k}$, representing the $k$-th state at the $j$-th step within the $i$-th subtree. Specifically, each state $s_{i,j,k}$ is characterized by the following length attributes:

1. **Incremental Length** ($\Delta L_{i,j,k}$): The number of tokens generated by $s_{i,j,k}$ during step $j$.

2. **Initial Context Length** ($L_{\text{init}}^{(i,j,k)}$): The total length of the prefix inherited from its ancestors:

$$L_{\text{init}}^{(i,j,k)} = L_{\text{in}} + \sum_{j' < j} \Delta L_{i,j',k_{j'}}, \tag{16}$$

where $k_{j'}$ denotes the index of the ancestor state at step $j'$ on the unique path to $s_{i,j,k}$.

3. **Final Context Length** ($L_{\text{final}}^{(i,j,k)}$): The total length after completing step $j$, defined as

$$L_{\text{final}}^{(i,j,k)} = L_{\text{init}}^{(i,j,k)} + \Delta L_{i,j,k}. \tag{17}$$

To maintain theoretical tractability while capturing the advanced characteristics of modern inference engines (e.g., vLLM), we establish the following assumptions:

**Assumption D.8 (Atomic Cost Components).** The inference cost of LLM is fundamentally decoupled into four atomic components. Given the batch size $b$, the model-specific architectural parameters including total parameter count $P$, number of layers $N_{\text{layer}}$, number of query heads $N_{\text{q}}$, number of KV heads $N_{\text{kv}}$, and head dimension $d_{\text{head}}$ (detailed configurations for evaluated models are provided in Table 11), and the storage precision $\text{prec}_{\text{p}}$ and $\text{prec}_{\text{kv}}$, we define the following components:

*Table 11.* Architectural parameters for the candidate models and verifiers, including the Qwen3 series (Yang et al., 2025) and the Skywork PRM series (He et al., 2024).

| Model Identifier | $P$ (B) | $N_{\text{layer}}$ | $N_{\text{q}}$ | $N_{\text{kv}}$ | $d_{\text{head}}$ | $\text{prec}_{\text{p}}$ | $\text{prec}_{\text{kv}}$ |
|---|---|---|---|---|---|---|---|
| *Candidate Models* | | | | | | | |
| Qwen3-0.6B | 0.75 | 28 | 16 | 8 | 128 | 2 | 2 |
| Qwen3-1.7B | 2.03 | 28 | 16 | 8 | 128 | 2 | 2 |
| Qwen3-4B | 4.02 | 36 | 32 | 8 | 128 | 2 | 2 |
| Qwen3-8B | 8.19 | 36 | 32 | 8 | 128 | 2 | 2 |
| Qwen3-14B | 14.77 | 40 | 40 | 8 | 128 | 2 | 2 |
| Qwen3-32B | 32.76 | 64 | 64 | 8 | 128 | 2 | 2 |
| *Verifiers* | | | | | | | |
| Skywork-o1-Open-PRM-Qwen-2.5-1.5B | 1.54 | 28 | 12 | 2 | 128 | 2 | 2 |
| Skywork-o1-Open-PRM-Qwen-2.5-7B | 7.61 | 28 | 28 | 4 | 128 | 2 | 2 |

1. **Parameter Computation ($f_{\text{p\_comp}}$):** The floating-point operations (FLOPs) required to process a single token through the linear layers:

$$f_{\text{p\_comp}}(b) = 2P \cdot b, \tag{18}$$

where the factor of 2 accounts for the *Fused Multiply-Add (FMA)* operation, representing one multiplication and one addition for each parameter per token.

2. **Parameter Memory Access ($f_{\text{p\_mem}}$):** The volume of data (Bytes) moved when loading the full model parameters from memory:

$$f_{\text{p\_mem}} = P \cdot \text{prec}_{\text{p}}. \tag{19}$$

3. **Attention Computation ($f_{\text{a\_comp}}$):** The FLOPs required for a single query token to compute attention scores and weighted sums against a context of length $l$:

$$f_{\text{a\_comp}}(b, l) = 4 \cdot b \cdot l \cdot N_{\text{layer}} \cdot N_{\text{q}} \cdot d_{\text{head}}, \tag{20}$$

where the factor of 4 accounts for two distinct matrix multiplication operations within the attention mechanism ($\mathbf{Q} \cdot \mathbf{K}^{\top}$ and $\mathbf{S} \cdot \mathbf{V}$), each contributing 2 FLOPs per element-wise dimension.

4. **Attention Memory Access ($f_{\text{a\_mem}}$):** The volume of data (Bytes) corresponding to the KV cache of length $l$:

$$f_{\text{a\_mem}}(b, l) = 2 \cdot b \cdot l \cdot N_{\text{layer}} \cdot N_{\text{kv}} \cdot d_{\text{head}} \cdot \text{prec}_{\text{kv}}, \tag{21}$$

where the factor of 2 accounts for Key Cache and Value Cache.

**Assumption D.9 (Advanced Engine Features).** We assume the inference engine supports the following advanced features:

1. **Prefix Sharing**: Multiple concurrent reasoning branches can logically share the same physical KV cache of their common prefix to minimize memory redundancy.

2. **Prefix Caching**: KV caches for common prefixes are automatically retained in memory and reused across discrete inference steps to avoid redundant computation.

3. **Dynamic Batching**: The engine supports real-time adjustment of the effective batch size as individual sequences within a reasoning step terminate at different lengths.

Note that the optimizations of prefix sharing and prefix caching are limited to generative architectures (e.g., model $M_t$), whereas they remain inapplicable to discriminative models such as the verifier.

Under the support of dynamic batching (Assumption D.9), we can now formally define the time-varying characteristics of tree-based decoding.

**Definition D.10 (Effective Batch Size $b_j(n)$).** In a step-synchronous reasoning step $j$, we define the effective batch size at token position $n$ as

$$b_j(n) = \sum_{i,k} \mathbf{1}[\Delta L_{i,j,k} \geq n]. \tag{22}$$

**Definition D.11 (Average Context Length $\bar{L}_j(n)$).** In step $j$, we define the average context length at token position $n$ as

$$\bar{L}_j(n) = \frac{1}{b_j(n)} \sum_{i,k} (L_{\text{init}}^{(i,j,k)} + n) \cdot \mathbf{1}[\Delta L_{i,j,k} \geq n]. \tag{23}$$

**Definition D.12 (Maximum Decoding Step $N_j$).** In step $j$, we define the maximum decoding step as

$$N_j = \max_{i,k} \Delta L_{i,j,k}. \tag{24}$$

Guided by the eFLOPs principle (Definition D.6), we derive the costs for each inference stage by identifying their unique operational characteristics.

**Proposition D.13 (Prefill Phase Cost).** *Given an input sequence of length $L_{\text{in}}$, the cost $C_{\text{prefill}}(1, L_{\text{in}})$ for the shared prefix processing is defined as:*

$$C_{\text{prefill}}(1, L_{\text{in}}) = L_{\text{in}} \cdot f_{\text{p\_comp}}(1) + \sum_{i=1}^{L_{\text{in}}} f_{\text{a\_comp}}(1, i) + (f_{\text{p\_mem}} + f_{\text{a\_mem}}(1, L_{\text{in}})) \cdot I \tag{25}$$

*Proof.* Under prefix sharing mechanism (Assumption D.9), the prefill phase processes the initial prompt as a single contiguous batch ($b = 1$).

- **Computation**: The linear projections are executed for all $L_{\text{in}}$ tokens ($L_{\text{in}} \cdot f_{\text{p\_comp}}(1)$). Due to the causal mask, attention FLOPs follow a discrete summation over the growing context: $\sum_{i=1}^{L_{\text{in}}} f_{\text{a\_comp}}(1, i)$.

- **Memory Access**: Model weights are loaded exactly once ($f_{\text{p\_mem}}$). The resulting KV cache for the entire prompt is then serialized to memory ($f_{\text{a\_mem}}(1, L_{\text{in}})$).

$\square$

**Proposition D.14 (Incremental Decoding Cost).** *The total decoding cost for step $j$ across $N_j$ token positions is:*

$$C_{\text{inc}}^{(j)} = \sum_{n=1}^{N_j} \left( f_{\text{p\_comp}}(b_j(n)) + f_{\text{a\_comp}}(b_j(n), \bar{L}_j(n)) + (f_{\text{p\_mem}} + f_{\text{a\_mem}}(b_j(n), \bar{L}_j(n))) \cdot I \right) \tag{26}$$

*Proof.* Enabled by prefix caching (Assumption D.9), decoding is performed as an incremental step-by-step process.

- **Computation**: At each position $n$, $b_j(n)$ new tokens are projected ($f_{\text{p\_comp}}$) and attend to their respective historical contexts of average length $\bar{L}_j(n)$ ($f_{\text{a\_comp}}$).

- **Memory Access**: Since decoding is memory-bound, each step $n$ incurs a mandatory reload of model weights ($f_{\text{p\_mem}}$) and a full retrieval of the active KV cache ($f_{\text{a\_mem}}$) from VRAM.

$\square$

**Proposition D.15 (Verification Cost).** *The verification cost at step $j$ is:*

$$C_{\text{ver}}^{(j)} = \sum_{i,k} \left( L_{\text{final}}^{(i,j,k)} \cdot f_{\text{p\_comp}}(1) + \sum_{n=1}^{L_{\text{final}}^{(i,j,k)}} f_{\text{a\_comp}}(1, n) \right) + I \cdot \left( f_{\text{p\_mem}} + \sum_{i,k} f_{\text{a\_mem}}(1, L_{\text{final}}^{(i,j,k)}) \right) \tag{27}$$

*Proof.* The verifier functions as a discriminative model requiring a complete forward pass over each sequence.

- **Computation**: For each branch $(i, k)$, the verifier computes the full log-likelihood of the final sequence $L_{\text{final}}^{(i,j,k)}$, treating it as a new prefill-style operation (Assumption D.9).

- **Memory Access**: Verifier weights are loaded once per verification task ($f_{\text{p\_mem}}$). However, since branches are evaluated as independent contexts in discriminative mode, the memory cost includes the total volume of KV data processed across all branches.

$\square$

*Remark* D.16. The total inference cost $C_{\text{UIS}}$ follows an additive decomposition:

$$C_{\text{UIS}} = C_{\text{prefill}}(L_{\text{in}}) + \sum_{j=1}^{H} \left[ C_{\text{inc}}^{(j)} + C_{\text{ver}}^{(j)} \right].$$

(28)

