# OpenReview forum: "UniScale: Adaptive Unified Inference Scaling via Online Joint Optimization of Model Routing and Test-Time Scaling"
_ICML.cc/2026/Conference — ICML 2026 regular_

### Official Review · Reviewer_p4iN · 2026-03-08

**Soundness:** 3
**Presentation:** 3
**Significance:** 3
**Originality:** 4
**Overall Recommendation:** 5
**Confidence:** 3

**Summary:**

This paper integrates model routing and test-time scaling into a single decision framework for optimizing the quality–cost trade-off in llm inference. Key innovations include path-aware early exiting, dense validator feedback using reward models, and a unified cost model.

**Compliance With Llm Reviewing Policy:**

Affirmed.

**Key Questions For Authors:**

See Weakness & Problems

**Limitations:**

yes

**Strengths And Weaknesses:**

Strength:
* Clear motivation. UIS breaks down boundaries between routing and TTS, creating a richer, continuous optimization space for inference control.

* Theoretically Grounded Framework. Formalizing configuration selection as a contextual bandit problem provides rigorous foundations for online adaptation under uncertainty.

* Thoughtful System Design. It proposes multiple practical mechanisms, e.g. path-aware early exit, address high-dimensional decision challenges.

Weakness & Problems:
* PRM quality. Could the authors provide more discussions on quantifying the model’s sensitivity to PRM errors, such as conducting additional evaluations with weaker verifiers?

* Limited evaluation scenarios. The empirical evaluation is narrowly focused on mathematical reasoning tasks (AIME 24/25, MATH-500) with 210 instances.It is recommended to expand the evaluation scenarios, for example, learning a policy on math datasets and then transferring it to coding tasks for validation.

* The unified reward function is a linear combination of three objectives, where the coefficients reflect the level of attention we place on model performance and computational cost. It is hoped that the authors can propose a quantitative method to guide the determination of these coefficients.

---

> ### Author Rebuttal · Authors · 2026-03-31
>
> We sincerely thank the reviewer for the assessment that our work is characterized by its clear motivation, theoretical soundness, and thoughtful design.
> ### W1. Verifier sensitivity
> We explicitly study the sensitivity to PRM quality by evaluating verifiers of different sizes (e.g., Skywork-o1-Open-PRM-Qwen-2.5-7B vs. Skywork-o1-Open-PRM-Qwen-2.5-1.5B). Increasing the PRM size (~5×) leads to different trade-offs: in the cost-sensitive mode (where LLM inference is lightweight), the additional verifier cost is more pronounced (cost +66%, accuracy +1.88pp); in the quality-priority mode (where LLM inference dominates), the cost increase is modest (cost +14%) while accuracy improves significantly (+6.25pp). These results show that while stronger PRMs can improve performance, UniScale remains effective even with weaker verifiers, indicating limited sensitivity to PRM quality and robustness to verification noise.
>
> Full results: https://anonymous.4open.science/r/ICML26_7872/verifier.png
> ### W2. Limited evaluation scenarios
> UniScale is inherently task-agnostic, as it only relies on proxy reward signals (e.g., PRM/RM) rather than task-specific correctness definitions. While current experiments focus on mathematical reasoning due to the availability of reliable evaluation signals, the framework itself directly extends to open-ended tasks where dense feedback can be provided by learned reward models. Importantly, UniScale learns a policy over semantic features rather than task-specific rules, which enables cross-task generalization. In particular, policies learned on math tasks can be transferred to other domains (e.g., coding) as long as compatible reward signals are available. Prior work [1] has demonstrated that RM-based approaches generalize to open-ended domains such as coding, QA and safety. Following the reviewer’s suggestion, we will further validate this by including coding benchmarks (e.g., LiveCodeBench) and studying cross-task transfer in the final version.
> ### W3. Reward weights
> The reward formulation in UniScale is designed to be flexible and orthogonal to weight selection: the coefficients reflect user preference over multiple objectives (e.g., accuracy vs. cost), and can be determined by standard multi-objective techniques or application-specific requirements.
>
> To better understand the impact of weight choices, we evaluate multiple configurations and construct accuracy–cost Pareto frontiers in the UIS space. UniScale consistently lies on the Pareto frontier and achieves broad coverage of the trade-off spectrum. Compared to baselines, it exhibits a steeper improvement trend, indicating more efficient configuration selection across different preference regimes. Moreover, UniScale remains stable across weight variations, with all variants lying on the Pareto frontier, suggesting that performance is robust to the specific choice of coefficients and avoids undesirable trade-offs. Building on these findings, we believe that a quantitative methodology for weight selection would enable a more effective leveraging of UniScale's potential, a direction we intend to explore in future work.
>
> Full results: https://anonymous.4open.science/r/ICML26_7872/pareto.png
>
> ---
> Thank you again for your detailed and insightful feedback. We hope that our response has addressed your concerns, but if we missed anything please let us know.
>
> [1] Ding, Dujian, et al. BEST-Route: Adaptive LLM Routing with Test-Time Optimal Compute. ICML'25

---

> > ### Author Rebuttal · Reviewer_p4iN · 2026-04-04
> >
> > My concerns have been addressed

---

> > > ### Author Response · Authors · 2026-04-04
> > >
> > > Thank you for your feedback. We are glad that our rebuttal has addressed your concerns. We appreciate your time and consideration.

---

### Official Review · Reviewer_g3mT · 2026-03-08

**Soundness:** 3
**Presentation:** 3
**Significance:** 3
**Originality:** 3
**Overall Recommendation:** 4
**Confidence:** 3

**Summary:**

The paper defines Unified Inference Scaling (UIS) which jointly optimizes model routing and test-time scaling (TTS) within a single decision space. Each inference configuration is a tuple (Model, QP, CP, BS), where QP controls the number of independent reasoning subtrees, CP the number of candidate expansions per step, and BS the beam size for pruning. The parameterization includes Chain-of-Thought, Best-of-N, Beam Search, and Diverse Verifier Tree Search as special cases.

The paper formulates adaptive configuration selection as a contextual multi-armed bandit and solves it with LinUCB. UniScale maps queries and configuration descriptions into a shared 1024-dimensional semantic space via a pretrained text embedding model, concatenates them into 2048-dimensional feature vectors, and learns a linear reward model online.

Besides, the work comes with three optimizations. 1. path-aware early exiting prunes unpromising search branches mid-inference, 2. dense verification feedback from a process reward model (PRM) supplements binary correctness with continuous quality signals, and 3. an FLOPs-based cost model unifies compute and memory costs.

Experiments on 210 math problems from AIME'24, AIME'25, and MATH-500, using six Qwen3 models (0.6B to 32B), show that UniScale achieves the highest composite reward and lowest cumulative regret compared to random, greedy, MLP, and k-NN baselines under both cost-sensitive and quality-priority reward modes.

**Compliance With Llm Reviewing Policy:**

Affirmed.

**Key Questions For Authors:**

- What is the standard deviation of accuracy and costs across different random seeds for Table 2?

- Can you provide accuracy vs. cost Pareto frontiers for each method? If not feasible, Reward computation is sensitive to the choice of weights. Shall we vary the weights to see how does the ranking of the methods in terms of reward change?

- Will the linear regression (for computing reward) overfit due to the limited samples?

**Limitations:**

Yes

**Strengths And Weaknesses:**

Strengths:

- The work introduces a unified formulation that combines model routing and test time scaling into a single action space. It capture the more dimensions that separate optimization would have missed. TTS smooths discrete performance gaps between models in the routing subspace. Model switching breaks the capacity ceiling of single-model TTS. The idea is neat.

- The action semantic representation is interesting. Converting configurations into structured text descriptions and embedding them with a pretrained model enables cross-action knowledge transfer. The ablation study confirms the benefits.

- I liked the evaluation on the non-stationary drifts. This is a practically important problem. And the solution implicitly handles the drifts by encouraging exploration.

Weaknesses:

- The evaluation dataset is quite small. It contains only 210 math problems. This is my main concern of the work. It is likely most configurations are visited at most once during the policy phase. Due to the small dataset, results in Table 2 could be sensitive to the warm-up sequences. As a comparison, RouterBench that is cited as baseline are evaluated on thousands of queries. If the work is on such a small dataset, providing some confidence intervals or variance across random seeds would be useful to justify the significance of the result.

- Besides, only math reasoning is tested. Both datasets has binary correctness. It is not clear to me how the approach would generalize to open-ended generation, coding or tasks where correctness is not binary. In particular, how would the dense verification feedback perform?

- For the ridge regression, the input is of dimension 2049 while the input samples are only less than 200. In this case, wouldn't we have an overfitting problem? since the input size is not big enough to sufficiently capture the distribution in such a huge feature space?

- Another concern is that UniScale does not dominate baselines on accuracy or cost individually, which are fundamental metrics here, rather than the reward. Looking at Table 2 does not intuitively show that UniScale is the parental frontiers. A figure similar to Figure 1 would be much better to interpret the result.

---

> ### Author Rebuttal · Authors · 2026-03-31
>
> We sincerely thank the reviewer for the assessment that our work is characterized by its conceptual novelty, design ingenuity, and practical significance.
> ### W1 & Q1. Data scale and stability
> To quantify variability, we extend the evaluation to 5 random seeds. UniScale achieves the best performance in 5/6 settings. Consistent with Table 2, UniScale only marginally trails Greedy in the small Routing space (6 actions), where low-dimensional constraints diminish the relative gains from exploration. Crucially, UniScale consistently exhibits lower standard deviations than Greedy, confirming that our exploration mechanism leads to stable convergence and effectively mitigates variance even with limited samples.
>
> Full results: https://anonymous.4open.science/r/ICML26_7872/performance.png
>
> While the dataset size is relatively small, UniScale targets an online decision-making setting, where sample efficiency is driven by feature generalization rather than dataset scale. In contextual bandits, each interaction provides immediate feedback to improve future decisions, making it fundamentally different from offline routing benchmarks such as RouterBench, which require large labeled datasets to train static predictors.
> ### W2. Generalization beyond math
> UniScale does not rely on binary correctness signals, although such signals can be incorporated as one component of the reward. More generally, our framework operates on proxy reward signals, which can be continuous and noisy, rather than exact or discrete. In particular, the dense verification feedback in UniScale is designed to provide fine-grained reward signals beyond binary correctness, making it naturally applicable to open-ended tasks. UniScale optimizes relative quality instead of absolute correctness, and dense feedback can be obtained from learned reward models or LLM-as-a-judge paradigms [1], which are widely used for open-ended generation, coding, and alignment tasks. This allows the framework to generalize to settings where exact correctness is unavailable. Prior work [2] has shown that TTS-style methods can extend beyond math reasoning to open-ended domains (e.g., QA, safety). We will further validate task generalization by including coding benchmarks (e.g., LiveCodeBench) and studying cross-task transfer in the final version.
> ### W3 & Q3. High-dimensional regression
> We use ridge regression, which is specifically designed for high-dimensional, small-sample settings through regularization, effectively controlling overfitting even when the feature dimension exceeds the number of samples. More importantly, our setting is an online contextual bandit rather than offline supervised learning. The linear model is used for reward estimation, and LinUCB updates parameters sequentially while explicitly modeling uncertainty to guide exploration. As a result, the model does not overfit to a fixed dataset but continuously refines its estimates through interaction. In addition, the feature space is not arbitrary high-dimensional noise but a semantically structured embedding, where features are highly informative and enable generalization across actions. This further alleviates the risk of overfitting despite the limited number of samples.
>
> Empirically, we do not observe overfitting: the regret curve (Fig. 7) shows stable convergence, and results are consistent across multiple random seeds with low variance (see W1 & Q1). These findings suggest that the current sample size is sufficient for reliable policy learning.
> ### W4 & Q2. Reward sensitivity / Pareto frontier
> Beyond the two original settings, we evaluate multiple reward weight configurations and explicitly construct accuracy–cost Pareto frontiers in the UIS space. The results show that UniScale not only lies on the Pareto frontier but also achieves broad coverage of the accuracy–cost trade-off spectrum. Specifically, under cost-sensitive mode, UniScale reduces computation to 48.2 TeFLOPs while maintaining competitive accuracy; under quality-priority mode, it achieves the highest accuracy (57.50%) with still competitive cost. Between these extremes, UniScale exhibits a steeper accuracy–cost improvement curve than baselines, indicating more efficient selection of UIS configurations. Moreover, all UniScale variants (across four weight settings) lie on the Pareto frontier, whereas other methods do not, suggesting that UniScale is robust to reward weight choices and avoids undesirable trade-offs (e.g., higher cost with lower accuracy).
>
> Full results: https://anonymous.4open.science/r/ICML26_7872/pareto.png
>
> ---
> Thank you again for your constructive feedback. We hope that our response has addressed your concerns, but if we missed anything please let us know.
>
> [1] Judging llm-as-a-judge with mt-bench and chatbot arena. NeurIPS'23
>
> [2] BEST-Route: Adaptive LLM Routing with Test-Time Optimal Compute. ICML'25

---

> > ### Author Rebuttal · Reviewer_g3mT · 2026-04-04
> >
> > I maintain the score of weak accept. The rebuttal addresses most of the questions. The work proposes some interesting ideas. The limitations the relatively narrow empirical study on math problems only. It is promised but not demonstrated.

---

> > > ### Author Response · Authors · 2026-04-04
> > >
> > > Thank you for your constructive feedback and for maintaining a positive assessment. We are glad that the rebuttal addressed most of your concerns.
> > >
> > > Regarding the scope of empirical evaluation, we agree that broader validation is important. As discussed in the rebuttal, our framework is not inherently limited to mathematical tasks and can extend to more open-ended settings with appropriate reward signals. We will further strengthen this aspect in the final version.
> > >
> > > We sincerely appreciate your insightful comments and suggestions.

---

### Official Review · Reviewer_c4mX · 2026-03-10

**Soundness:** 3
**Presentation:** 3
**Significance:** 3
**Originality:** 2
**Overall Recommendation:** 4
**Confidence:** 4

**Summary:**

This paper introduces Unified Inference Scaling (UIS), which combines model routing and test-time scaling (TTS) into a single joint decision space parameterized by (M, QP, CP, BS). The authors use LinUCB, a contextual multi-armed bandit algorithm, to adaptively select inference configurations online. The framework includes four auxiliary components: action semantic representations via text embeddings, path-aware early exiting, dense verification feedback, and an eFLOPs-based UIS cost model. Experiments are conducted on math reasoning benchmarks (AIME'24, AIME'25, MATH-500) using the Qwen3 model family.

**Compliance With Llm Reviewing Policy:**

Affirmed.

**Key Questions For Authors:**

as mentioned in the weakness section

**Limitations:**

above

**Strengths And Weaknesses:**

## Strengths

1. Unifying model routing and TTS into a single decision space (M, QP, CP, BS) fills a real gap. TTS smooths routing's discrete jumps; routing breaks single-model TTS ceilings.

2. Contextual bandits naturally fit non-stationary deployment. The drift experiments (model add/remove, reward mode switch) convincingly demonstrate adaptability.

3. The eFLOPs cost model (Appendix D.3) carefully accounts for prefill, decoding, verification, prefix sharing, and dynamic batching.

4. All major components (semantic representations, dense feedback, early exiting, α) are individually ablated.


## Weaknesses

1. Only 210 queries (160 policy-driven), no confidence intervals or significance tests. In a 1024-d feature space, this is far too small for meaningful bandit convergence .

2. Math-only evaluation (AIME, MATH-500). No code, QA, instruction following, or open-ended tasks. Claims of a general framework are unsupported.

3. No theoretical regret bound despite being a bandit paper. The confidence ellipsoid in Appendix D.1 is never extended to an actual guarantee.

4. Greedy (α=0) is competitive in many settings, suggesting the linear reward assumption captures only obvious patterns and exploration adds limited value.

5. The framework hinges on a PRM verifier with AUROC of only 0.8067. No robustness analysis to verifier quality; PRM verifiers are inapplicable to open-ended tasks.

6. Missing Thompson Sampling, Neural UCB, and crucially BEST-Route which is the most directly related work, cited but not compared.

7. Hand-crafted text templates embedded by a generic encoder is indirect and fragile. No template sensitivity analysis.

8. Qwen3 only. No cross-family (Llama, Mistral, DeepSeek) evaluation.

9. Encoder inference, matrix inversion (d=1024), and per-step verifier calls are never included in cost comparisons.

10. Reward weights w₁, w₂, w₃ are manually set with no sensitivity analysis or principled multi-objective approach.

---

> ### Author Rebuttal · Authors · 2026-03-31
>
> We sincerely thank the reviewer for the assessment that our work is characterized by its conceptual novelty, dynamic adaptability, modeling rigorousness, and empirical thoroughness.
> ### Q1 Sample size and convergence
> Although the query number is limited, UniScale operates in a semantic embedding space, enabling generalization with fewer samples. We report results over 5 seeds: UniScale achieves the best performance in 5/6 settings while exhibiting lower variance than Greedy, indicating stable convergence.
>
> Results: https://anonymous.4open.science/r/ICML26_7872/performance.png
> ### Q2 Task generalization
> UniScale only requires proxy rewards (PRM/RM) and is task-agnostic. While existing PRMs and most TTS studies focus on math reasoning [1], [2] shows that RM-based approaches extend to open-ended tasks (QA, safety). We will include coding benchmarks (e.g., LiveCodeBench) in the final version.
> ### Q3 Regret bound
> UniScale follows LinUCB and inherits its regret guarantees under linear reward assumptions. Our formulation matches standard linear bandits with learned embeddings; empirical regret (Fig.7) shows stable convergence. We will include explicit discussion in future work.
> ### Q4 Exploration vs. Greedy
> As discussed in Sec. 4.2, Greedy is competitive in the small routing space (6 actions) due to limited complexity. However, in the larger UIS space (168 actions), it significantly underperforms UniScale, showing that exploration is essential for discovering non-obvious high-quality configurations. Moreover, exploration guarantees stable convergence with low variance (see Q1).
> ### Q5 PRM quality and applicability
> (1) Roubustness: UniScale optimizes relative rewards and does not require perfect PRM. We study the impact of verifiers using Skywork-o1-Open-PRM-Qwen-2.5-7B. Scaling verifier size (~5×) leads to different trade-offs: in the cost-sensitive mode (LLM inference is lightweight), the additional verifier cost is more pronounced (cost +66%, accuracy +1.88pp); in the quality-priority mode (LLM inference is expensive), the cost increase is modest (+14%) while accuracy improves significantly (+6.25pp). These results show that UniScale benefits from stronger verifiers but does not critically depend on them.
>
> Results: https://anonymous.4open.science/r/ICML26_7872/verifier.png
>
> (2) Applicability: UniScale relies only on proxy rewards and can extend to open-ended tasks with learned reward models.
> ### Q6 Baselines
> We add NeuralUCB and Thompson Sampling; UniScale consistently outperforms both (UIS space, quality-priority mode: +7.12pp vs NeuralUCB at 76% cost; +3.37pp vs TS at 73% cost).
>
> Results: https://anonymous.4open.science/r/ICML26_7872/performance.png
>
> BEST-Route's space has a significantly lower upper bound than UIS (cost-sensitive mode: +14.37pp at 10% cost). Notably, UniScale also performs effectively when exploring within the BEST-Route space.
>
> Results: https://anonymous.4open.science/r/ICML26_7872/BEST-Route.png
> ### Q7 Template robustness
> We use structured key-value templates encoding general attributes, which are more stable than templates in [3]. Since the representation relies on structured attributes rather than wording, it is robust to minor template variations.
> ### Q8 Model family generalization
> UniScale is model-agnostic and only depends on high-level specifications (Appendix B.1). We will include cross-family evaluations (e.g., LLaMA) in the final version.
> ### Q9 Computational overhead
> We explicitly account for all omitted components in the cost breakdown. Query embedding (encoder inference) accounts for 0.77%, bandit overhead for 0.14% (we use the Sherman-Morrison formula for rank-1 updates, which requires only $O(d^2)$ operations, rather than explicit $O(d^3)$ matrix inversion), and per-step PRM verification for 8.02% of total time. All three are included in our measurement and remain small compared to LLM Inference.
>
> Results: https://anonymous.4open.science/r/ICML26_7872/overhead.png
> ### Q10 Reward weights
> We evaluate multiple reward weight configurations and construct accuracy–cost Pareto frontiers in the UIS space. UniScale consistently lies on the Pareto frontier and achieves broad coverage trade-off. Compared to baselines, UniScale exhibits a steeper improvement trend, indicating more efficient selection of configurations. Moreover, its performance remains stable across different weight settings, with all variants lying on the Pareto frontier, demonstrating robustness to reward weights.
>
> Results: https://anonymous.4open.science/r/ICML26_7872/pareto.png
>
> ---
> Thank you again for your encouraging and insightful feedback. We hope that our response has addressed your concerns, but if we missed anything please let us know.
>
> [1] Scaling LLM test-time compute optimally can be more effective than scaling parameters for reasoning. ICLR'25
>
> [2] BEST-Route: Adaptive LLM Routing with Test-Time Optimal Compute. ICML'25
>
> [3] Masrouter: Learning to route llms for multi-agent systems. ACL'25

---

> > ### Author Rebuttal · Reviewer_c4mX · 2026-04-03
> >
> > Thank you for your reply. You have solved all of my concerns!

---

> > > ### Author Response · Authors · 2026-04-04
> > >
> > > Thank you very much for your positive feedback. We are glad that our rebuttal has addressed your concerns. We appreciate your time and thoughtful evaluation.

---

### Official Review · Reviewer_49Mm · 2026-03-10

**Soundness:** 2
**Presentation:** 3
**Significance:** 2
**Originality:** 2
**Overall Recommendation:** 2
**Confidence:** 3

**Summary:**

This paper introduces Unified Inference Scaling (UIS), a formulation that jointly optimizes model routing and test-time scaling (TTS) in a single decision space. The authors then propose UNISCALE, an online framework that formulates UIS configuration selection as a contextual multi-armed bandit problem, solved via LinUCB. The paper further adds three practical improvements specific to the problem context. Experiments on math benchmarks using the Qwen3 model family demonstrate improvements over baselines in quality and cost efficiency.

**Compliance With Llm Reviewing Policy:**

Affirmed.

**Key Questions For Authors:**

Detailed Comments to the Authors
- All experiments are conducted on mathematical reasoning tasks (e.g., AIME) using a single model family (Qwen3). This raises significant concerns about the proposed solution’s generalizability. Math problems are known to have well-defined correctness definitions that make it easy for verification. How does UNISCALE perform on other open-ended tasks where verification is noisy or unavailable? Also, it would be beneficial to see its performance on other model families.

- It’s unclear to me what the actual size of the action space is used in the paper and the experiments. Appendix D.2 reports 168 configurations, but it seems inconsistent with the configuration range in Table 1. With 6 models * multiple TTS settings, possible number of configurations derived from Table 1 seems much greater than 168. Please more explicitly calculate and state the action space size.

- This brings another key concern: LinUCB calculation requires matrix inversion in each step which is of O(d^3) complexity. It seems like the matrix dimension is 2048 when concatenating two1024-d embeddings, which seems cost-prohibitive for doing inversion in every step. This computational overhead of the bandit itself is not sufficiently discussed in the paper. How does the overhead compare to the inference cost itself is unclear. If scalability is a central selling point, I would expect experiments that increase number of models, number of TTS settings, feature dimensionality, and report decision overhead.
The 50-step warm-up phase seems short for such a high-dimensional space. How sensitive is performance to the warm-up length?

- The baselines used (random and greedy for bandit, MLP/k-NN for predictive) are relatively simple. There are tons of online bandit or bayesian optimization approaches proposed in recent years. The paper should compare against these more competitive approaches in the literature. Several methods are cited in the related work, but not compared against either.
How is the Oracle baseline computed? Is it enumerating over all possible configurations for each query?

Please explain.
How were the reward weights in Eq. 4 chosen? Was any sensitivity analysis conducted beyond the two fixed modes?

- The core assumption that the rewards are linear is strong but may not capture reality. For example, aggressive TTS on a very small model may result in qualitatively different returns than on a larger model. While the authors argue that the Transformer encoder captures non-linearity during feature extraction, I am not fully convinced since the encoder is not fine-tuned per task, and the semantic descriptions are handmade templates that may not transfer across domains.

- Results in Table 2 do not report standard deviations or confidence intervals. With only 210 test instances and 160 iterations, I would expect the variance to be substantial.

- The solution seems to heavily rely on the verifier for both dense feedback and path evaluation. Though, the computational cost of the verifier itself is unclear. The effect of the verifier should be isolated in the ablation study. Also, I would assume the performance of UNISCALE would directly depend on the quality of the verifier, but no empirical evidence is provided to give more insight into this correlation.

- Overall, I like the idea of unifying the LLM inference challenges and modelling it as a bandit problem. However, the limited evaluation and missing analysis of key assumptions and settings make me unable to recommend acceptance in its current form.

**Limitations:**

Yes, could be improved.

**Strengths And Weaknesses:**

Strong Points
- The paper is well-motivated. Unifying model routing and TTS is a natural idea that may lead to improvement in balancing LLM inference quality and cost compared to handling them separately.
Modelling the problem as a contextual bandit and applying LinUCB is a reasonable and theoretically sound choice.
- The authors make a good effort in formally discussing the theoretical aspects of the algorithm.
Multiple deployment scenarios are experimented along with ablation studies for key components of the proposed solution.
- The paper is generally easy to follow and well-written.

Weak Points
- Experiments are limited in task scope and model selection.
- Baselines used are limited and not sufficiently strong to compare against.
- Scalability demonstrated does not seem consistent and raises concerns.
- Some assumptions and settings are not thoroughly explained.

The above concerns are detailed below.

---

> ### Author Rebuttal · Authors · 2026-03-31
>
> We sincerely thank the reviewer for the assessment that our work is well-motivated, theoretically sound, and well-written, and for recognizing our experiments across multiple deployment scenarios and ablation studies.
> ### Q1 Generalizability
> (1) Task. UniScale only requires proxy rewards (PRM/RM) and does not rely on exact verification. Thus it is inherently task-agnostic and applicable to open-ended tasks. While current evaluation follows prior TTS work on math [1], [2] shows such approaches extend to QA and safety. We will include coding benchmarks (e.g., LiveCodeBench) in the final version.
>
> (2) Model. UniScale is model-agnostic, as it only depends on high-level model specifications (Appendix B.1). We will include cross-family evaluation (e.g., LLaMA) in the final version.
> ### Q2 Action space size
> We clarify that Table 1 constrains $QP, CP \in \\{1, 2, 4, 8, 16, 32, 64\\}$ with $QP \cdot CP \leq 64$, resulting in 28 valid TTS configurations per model.
> ### Q3 Overhead & warm-up
> (1) We avoid $O(d^3)$ inversion via Sherman–Morrison, reducing to $O(d^2)$. Empirically, the total decision overhead (query embedding + bandit overhead) is 0.91% of total time, which is negligible compared to LLM inference. While scalability is not our primary focus, the negligible overhead observed across 168 actions demonstrates the efficiency of UniScale in a substantial decision space.
>
> Results: https://anonymous.4open.science/r/ICML26_7872/overhead.png
>
> (2) Despite high dimensionality, semantic embeddings enable efficient generalization. Regret curves (Fig.7) show 50-step warm-up is sufficient. This aligns with [3], which uses higher dimension (5120) with shorter warm-up (40).
> ### Q4 Baselines
> We add NeuralUCB [4] and Thompson Sampling [5]; UniScale consistently outperforms both (UIS space, quality-priority mode: +7.12pp vs NeuralUCB at 76% cost; +3.37pp vs TS at 73% cost).
>
> Results: https://anonymous.4open.science/r/ICML26_7872/performance.png
>
> Oracle selects the optimal configuration per query via exhaustive enumeration.
> ### Q5 Reward weights
> We use reward weights to instantiate different trade-offs (e.g., cost-sensitive vs. quality-priority). We evaluate multiple reward weight configurations and construct accuracy–cost Pareto frontiers in the UIS space. UniScale consistently lies on the Pareto frontier and achieves broad coverage trade-off. Compared to baselines, UniScale exhibits a steeper improvement trend, indicating more efficient selection of configurations. Moreover, its performance remains stable across different weight settings, with all variants lying on the Pareto frontier, demonstrating robustness to reward weights.
>
> Results: https://anonymous.4open.science/r/ICML26_7872/pareto.png
> ### Q6 Linearity assumption
> We do not assume linearity in the raw configuration space. Instead, structured templates and pretrained encoders map configurations into a semantic space where linear models approximate reward effectively. Empirically, UniScale outperforms NeuralUCB in all scenarios (see Q4), suggesting that the linear approximation is sufficient in practice.
> ### Q7 Variance
> We extend the evaluation to 5 random seeds and report the full results. UniScale achieves the best performance in 5/6 settings, while exhibiting consistently lower standard deviations compared to Greedy. This indicates that the exploration mechanism leads to stable convergence and effectively mitigates variance.
>
> Results: https://anonymous.4open.science/r/ICML26_7872/performance.png
> ### Q8 Verifier cost and quality
> (1) Cost. We explicitly measure verifier overhead: it accounts for 8.02% of total time (quality-priority mode), compared to 88.02% for LLM inference, and thus does not dominate system cost.
>
> Results: https://anonymous.4open.science/r/ICML26_7872/overhead.png
>
> (2) Quality. We study the impact of verifier quality using Skywork-o1-Open-PRM-Qwen-2.5-7B. Increasing the PRM size (~5×) leads to different trade-offs: in the cost-sensitive mode (LLM inference is lightweight), the additional verifier cost is more pronounced (cost +66%, accuracy +1.88pp); in the quality-priority mode (LLM inference is expensive), the cost increase is modest (+14%) while accuracy improves significantly (+6.25pp). These results show that UniScale benefits from stronger verifiers but does not critically depend on them.
>
> Results: https://anonymous.4open.science/r/ICML26_7872/verifier.png
>
> ---
> Thank you again for your time and your careful feedback. We hope our clarifications and additional results could improve your opinion of our work.
>
> [1] Scaling LLM test-time compute optimally can be more effective than scaling parameters for reasoning. ICLR'25
>
> [2] BEST-Route: Adaptive LLM Routing with Test-Time Optimal Compute. ICML'25
>
> [3] Use Your INSTINCT: INSTruction optimization for LLMs usIng Neural bandits Coupled with Transformers. ICML'24
>
> [4] Neural contextual bandits with ucb-based exploration. ICML'20
>
> [5] An empirical evaluation of thompson sampling. NIPS'11

---

### Decision · Program_Chairs · 2026-04-30

**Decision:**

Accept (regular)

**Comment:**

In this paper, the authors propose UniScale which frames routing and test time scaling as a single decision problem over a combined action space. The observation that routing and test time scaling share the same action structure motivates this treatment. This subsumes CoT, Best-of-N, Beam Search, Tree search as special cases. Configuration (or joint action) selection is cast as a contextual bandit solved with LinUCB over embeddings from a transformer encoder. The observation and insights stem from practical framing and reviewers are split 5/4/4/2 after the rebuttal. The reviewer concerns were marked resovled after extensive experiements (NeuralUCB, Thompson sampling, various routing and pareto frontier analysis). The evaluation is narrow on 210 math instances with a single Qwen3 model family. Although the gains report on cost and quality are real, they are not strong enough to carry a math-only, single-family evaluation. For these reasons, I am leaning towards weak accept.